# Anion Exchange Membranes for Fuel Cell Application: A Review

**DOI:** 10.3390/polym14061197

**Published:** 2022-03-16

**Authors:** Gautam Das, Ji-Hyeok Choi, Phan Khanh Thinh Nguyen, Dong-Joo Kim, Young Soo Yoon

**Affiliations:** 1Department of Polymer Science and Engineering, School of Applied Chemical Engineering, Kyungpook National University, Daegu 41566, Korea; gautam2706@gmail.com; 2Department of Materials Science and Engineering, Gachon University, Seongnam 13120, Gyeonggi-do, Korea; joshua9456@gachon.ac.kr; 3Department of Chemical and Biological Engineering, Gachon University, Seongnam 13120, Korea; thinhnpk@gachon.ac.kr; 4Materials Research and Education Center, Auburn University, 275 Wilmore Labs, Auburn, AL 36849, USA

**Keywords:** alkali stability, ionic conductivity, power density, non-platinum catalyst, low cost, anion exchange membrane, fuel cell, renewable energy

## Abstract

The fuel cell industry is the most promising industry in terms of the advancement of clean and safe technologies for sustainable energy generation. The polymer electrolyte membrane fuel cell is divided into two parts: anion exchange membrane fuel cells (AEMFCs) and proton exchange membrane fuel cells (PEMFCs). In the case of PEMFCs, high-power density was secured and research and development for commercialization have made significant progress. However, there are technical limitations and high-cost issues for the use of precious metal catalysts including Pt, the durability of catalysts, bipolar plates, and membranes, and the use of hydrogen to ensure system stability. On the contrary, AEMFCs have been used as low-platinum or non-platinum catalysts and have a low activation energy of oxygen reduction reaction, so many studies have been conducted to find alternatives to overcome the problems of PEMFCs in the last decade. At the core of ensuring the power density of AEMFCs is the anion exchange membrane (AEM) which is less durable and less conductive than the cation exchange membrane. AEMFCs are a promising technology that can solve the high-cost problem of PEMFCs that have reached technological saturation and overcome technical limitations. This review focuses on the various aspects of AEMs for AEMFCs application.

## 1. Introduction

Greenhouse gases generated by fossil-based fuels have escalated global warming. Furthermore, the rapid depletion of oil and natural gas reserves poses a serious issue for the future. Among the various potential alternative energy options, fuel cells offer the most promising solution [1]. Other alternative sources of energy generation such as wind and solar are dependent on weather conditions and require a large factorial setup which can increase the cost of operation. On the other hand, fuel cells, unlike other promising renewable energies (e.g., solar, wind, hydropower, etc.), can continuously generate electricity as long as the fuel injection continues. This advantage can generate greater synergy if it can be appropriately used for the power grid by linking it to an energy storage system to reduce the spatial and temporal gap with intermittent energy sources [2]. Therefore, fuel cells are one of the most researched topics in materials science because they can deal with problems related to the global energy crisis and the environmental shortcomings associated with petroleum-based materials [3]. Fuel cell technologies offer the advantage of the direct conversion of chemical energy into electrical energy with the generation of heat and water [4]. Fuel cells exhibit an efficiency as high as 60% and with >90% reduction in harmful pollutants [5]. Similarly, sustainable energy with environmental beingness is also realized through secondary batteries. As shown in Table 1, batteries have advantages such as power density and a life cycle depending on the type [6], but when applied in transportation, a long-distance operation is impossible due to technical limitations such as their storage and charging time [7,8].

Currently, the academic and industrial communities are aiming to build a hydrogen economy by adopting greener technologies for hydrogen production and infrastructure construction, aiming to achieve zero carbon across all stages from production to consumption [10,11]. Therefore, a strong synergistic effect can be created when fuel cells are used in conjunction for zero emission. The advantage of fuel cells is that they generate a higher energy density than batteries [12,13]. Thus, fuel cells represent an emerging class of energy devices that can compensate for the problems of batteries such as storage limits, and when used for transportation, they have the advantage of increasing their mileage and power responsiveness [14]. The major applications of fuel cells are in the transportation industry, primarily owing to their controlled emission of greenhouse gases. Other potential applications include stationary/distributed and portable power generation [15]. PEMFCs are a commercially available technology that is well-established and attracts attention due to their high ionic conductivity, chemical, mechanical and thermal stability, and high durability under humid conditions. A number of companies, such as Hyundai Motors, Nikola, General Motors, and Toyota have utilized fuel cells to power light vehicles and have plans to develop and commercialize these vehicles during the period 2020–2030 [15,16,17].

However, owing to the high cost of platinum group metal (PGM)-based catalysts, the requirements of high corrosion resistance metal bipolar plates, cumbersome maintenance issues, and the unavailability of PEM with ionic conductivity, as is the case of perfluorosulfonic acid (PFSA) membranes such as Nafion membranes [16,18,19], have limited the widespread use of PEMFCs as an affordable energy generation device. Thus, research on AEMFCs is continuously increasing as an alternative to these problems. The use of AEMs under alkaline conditions enables faster electrochemical activity to be achieved, reducing the amount of non-precious and noble metals used and reducing fuel crossover [19,20]. AEMs are the core technology that determines AEMFC performance [21,22]. For commercialization, ionic conductivity and durability in an alkaline environment must be ensured. This review discusses the various backbones (polyvinyl alcohol, polysulfones, polyphenylene, ionic liquid, poly-olefin-based membranes) including ionomer and membrane design, as the basic steps to study AEM fundamentals and applications.

## 2. Overview of Fuel Cells

### 2.1. History

First invented by William Grove in 1839, fuel cells have been in development for 180 years [23]. Over the years, fuel cell technologies have undergone several cycles of intense research activity. Today, they are viewed as efficient energy conversion devices with high power efficiencies, ease of operation, and low emissions. The utilization of fuel cells as energy devices was first realized by General Electric Co. (GE) for NASA’s Gemini (1963) and Apollo (1968) space programs [24]. However, the use of liquid aqueous electrolytes made this technology vulnerable to poisoning by carbon dioxide which leads to the formation of carbonate precipitates.

The development of the AEM for alkaline fuel cells resulted in a drastic increase in fuel-cell popularity among the scientific and industrial communities [25,26,27]. This technology is appealing on the basis of cost, as most fuel cells use non-noble metals and fast electrokinetics in the alkaline medium. Furthermore, the absence of the liquid electrolyte (aqueous KOH) prevents electrode weeping and corrosion. By comparison, the ionic resistivity of the electrolyte is lower in the case of the AEMFC (0.05 vs. 0.08 ohm cm^−2^ for the PEMFC). However, despite the advent of next-generation ultra-thin acidic polymer membranes, AEMFC’s intrinsic advantages, such as good cathode kinetics and electrode polarization, surpass those of the PEMFC.

### 2.2. Principle of Fuel Cell Operation

Polymer electrolyte membrane fuel cells use the membrane as a cell separator and electrolyte and are capable of generating high energy densities, which is conducive to portable applications. The prime advantage of this technology is that it eliminates the problem of handling highly corrosive chemicals. The operating temperature for this technology is in the range of 60–80 °C, thereby minimizing operational costs.

The membrane plays an important role in conducting the ions to the cathode, where they combine with oxygen to form water. Thus, the appropriate choice of membrane is important for maximizing the performance of the fuel cell. Generally, the high conductivity of the membrane is essential in order to compensate for resistive loss and ionic resistivity. Most of the present technologies utilize ion-conducting polymers as an electrolyte in PEMFCs. The distribution and strength of the ionic groups strongly influence membrane performance [28]. Proton transport through an aqueous Nafion membrane has been investigated by many researchers using various methods, including ab initio quantum chemistry, molecular dynamics, Poisson–Boltzmann theory, non-equilibrium statistical mechanical modeling, and dielectric saturation [29,30,31,32,33].

### 2.3. Anion Exchange Membrane Fuel Cells (AEMFCs)

#### Background

The alkaline fuel cell (AFC) is an electrochemical device that converts chemical energy into electrical energy and employs an aqueous solution of potassium hydroxide (KOH), the most conductive of all alkali metal hydroxides, as the electrolyte [34]. AFCs are the oldest type of fuel cell and have been employed for the Gemini space program by NASA [35,36]. In 1959, Ihrig and co-workers designed an alkaline fuel cell of 15 kW containing KOH as the electrolyte and used it to run a 20 hp tractor. The charge carrier in AFC is the OH^−^ ion rather than a proton, and thus the electro-oxidation of the fuel on the anode results in the generation of H^+^, which reacts with the OH^−^ to produce water; the electrons are carried through an external load. The electrons then combine with oxygen and water at the cathode, generating OH^−^. The overall reaction can be summarized as follows:Anode reaction: 2H_2_ + 4OH^−^ → 4H_2_O + 4e^−^
(1)
Cathode reaction: O_2_ + 2H_2_O + 4e^−^→ 4OH^−^(2)
Overall cell reaction: 2H_2_ + O_2_ → 2H_2_O + electrical energy + heat(3)

The prime advantages of the AFC is its low operating temperature requirements (60–70 °C) and a less harsh environment compared to those of the PEMFC. Furthermore, the reaction kinetics are faster in the basic medium than under acidic conditions, leading to higher cell output voltages. Another important aspect of the AFC is the high-pH conditions which result in low overpotentials in the electrochemical reaction and thus allow the use of non-noble-metal catalysts such as nickel and its alloy, silver, and cobalt [37,38,39]. Furthermore, AFCs offer fuel flexibility, offer better reaction kinetics, and drastically reduce fuel crossover [34,40]. As shown in Figure 1, AFC technology utilizes a liquid electrolyte for ionic conduction between two electrodes. Potassium hydroxide is the most commonly used electrolyte for this purpose because it is the most conductive of all alkali metal hydroxides. All these advantages make AFC technology financially and technically feasible.

AEMFC technologies gained widespread recognition after Cheng et al. [41] proposed the possibility of using a polymer electrolyte membrane instead of a conventional liquid electrolyte. Over the years, research on the development of solid polymer electrolyte membranes has increased tremendously [42,43]. The introduction of AEM in alkaline fuel cells (AEMFCs) has largely alleviated the problems associated with the liquid electrolytes. Solid electrolytes are a recent development in the ongoing effort to obtain a high-performing, industrially feasible energy device. The use of the solid membrane minimizes the carbonation and fuel crossover issue associated with aqueous KOH [44], and the membrane possesses excellent chemical, thermal, and mechanical properties. The alkaline exchange membrane and the electrodes form the heart of the fuel cell. The selectivity and conductivity of the OH^−^ ions through the membrane are a prerequisite for an efficient fuel cell. However, the choice of membrane and its subsequent design requires innovative efforts. The membrane should act as a barrier to the fuel, be stable in the medium used, maintain its durability regardless of the hydration state, and retain its properties at the working temperature [37]. The overall reaction in the case of AFCs can be summarized by Equation (4): 2H_2_ + O_2_ → 2H_2_O(4)

The reaction is accompanied by the generation of heat and electrical energy. The fuel (mainly hydrogen, H_2_) is fed onto the anode where it is oxidized and reacts with hydroxide (OH^−^) to form a water molecule; the electron generated in the process travels through an external circuit to the cathode. However, as shown in Table 2 the ionic conductivity of AEM is generally lower than the commercial-grade proton exchange membrane (PEM).

Moreover, the degradation rate of the membrane used in PEMFCs is much lower than that of the membrane used in AEMFCs owing to the highly corrosive environment of the latter. However, the biggest advantage of the alkaline fuel cell is that there is minimal fuel crossover. The biggest contributing factor for fuel crossover is the electroosmotic drag and concentration difference between the anode and cathode. In the case of PEMFC, the H^+^ ion and fuel flows in the same direction and is responsible for the fuel cross-over problems, whereas in the case of AEMFCs, the fuel crossover occurs opposite to the direction of the movement of the OH^−^ ion (Figure 2) which greatly reduces the electroosmotic drag. AEMFCs drew attention following the use of this technology in automobiles by K. Kordesh around 1970 [48]. Despite such success, AEMFCs suffered initial setbacks owing to the high material cost and certain shortcomings in the operation of the electrochemical devices [49]. However, this research area has attracted an important amount of attention in recent years, and the use of solid electrolyte membranes has significantly increased the efficiency of this type of cell.

### 2.4. Challenges in AEMFC Operations

#### 2.4.1. Ionomers

Unlike the proton exchange membrane fuel cells (PEMFCs), which have been studied with a great amount of research investment, AEMFCs lack standard commercially available materials and standard baseline procedures (test conditions and properties). The ionomer is an important component of AEMFCs which forms an ion transport pathway between the membrane and the reaction sites in the catalysts layer. However, it has been a challenging feat to develop highly conductive and stable anion exchange ionomers (AEI). Few researchers have tried to apply Nafion-based ionomers in constructing membrane electrode assemble (MEA) for AEMFC application [50,51,52]. However, Nafion is proton conducting and thus, its applicability in AEMFC which relies on OH^−^ conduction has found limited use. The absence of a suitable ionomer, as in the case of PEMFCs, has been a bottleneck in the enhancement of the power density of AEMFCs. It is to be noted that in improving the performance of AEMFCs, ionomers play an intrinsically important part, for instance, the AEI is used to provide a robust hydroxide conduction channel for the electrode and thus establish a three-phase boundary (TPB) where the electrochemical reaction takes place. Matsuoka et al. [53] fabricated an MEA using quaternized poly(4-vinyl pyridine) as the ionomer for alkaline direct alcohol fuel cells. They suggested that excessive water uptake by the ionomer in the electrodes results in performance loss as a result of the blockage of the active catalytic sites. Quaternized fluorinated copoly(arylene ether) ionomers were subsequently reported by Zhou et al. [54] which exhibited greater hydrophobicity owing to the presence of octafluoro-biphenyl groups in the backbone (Figure 3). Unlu et al. [55] fabricated an anion exchange ionomer (AEI) functionalized with a quaternary ammonium group stored at 5 wt.% (DMF). The electrodes were prepared after casting a mixture of Pt/C catalyst (20%) with the 5 wt.% AEI in DMF (weight ratio of 2:3 to water). Matsumoto et al. [56] designed a catalyst ink with KOH-doped polybenzimidazole (PBI) and platinum nanoparticles deposited on carbon nanotubes (CNTs). The AEI induces excellent mass transfer at the catalyst interface, resulting in a power density of 256 mW cm^−2^ under H_2_/air conditions. They stated that the addition of the same AEI to the anode and cathode catalyst layers decreases the interfacial resistance of the MEA, thereby improving the performance.

Yang et al. [57] performed a mass-specific optimization study of the ionomer using a 5 wt.% As-4 solution (Tokuyama Corporation, Tokyo, Japan). A power density of 358 mW cm^−2^ at 20 wt.% and H_2_/O_2_ (100% relative humidity, RH) was obtained by the authors using a single cell setup. The author reported that the continuous ionomer network in the pore structure of the catalysts layer provides conduction diffusion channels, thereby enhancing the catalyst efficiency. Generally, the formation of a proper TPB formation is essential for maximum catalyst utilization. It should be understood that too much ionomer content results in the blockage of the surface pores and the shielding of the surface of the electrode, thereby decreasing the efficiency of the catalysts’ utilization. Thus, the utilization of ionomers should be such that it uniformly coats the catalyst but does not block the gas diffusion. Ahlfield et al. [58] studied the influence of ionomers with different molecular weights on the performance of AEMFC. They observed that the use of low-molecular-weight ionomers improved fuel cell performance. Furthermore, they indicated that the promotion of systematic phase separation through the introduction of hydrophobic blocks would create ion conduction and gas diffusion channels. Liang et al. [59] revealed the importance of the shape of the shared network shell surrounding the Pt/C nature of the ionomer in the optimization of electrode design. A hydrophilic ionomer may adsorb and desorb a large amount of water, which may interfere with the flooding of the electrode and gas transport properties. Considering that ionomers in a hydrated state can degrade the AEMFC performance by generating isolated aggregates due to excessive expansion, the ionomer crosslinking immobilization strategy can alleviate the above problem, enabling the efficient generation and transport of electrons. In their work, Huang et al. [60] used an AEI comprising of a composite structure of poly(norbornene)-reinforced block copolymer with pendant N,N,N′, N′-Tetramethyl-1,6-hexanediamine (TMHDA) as quaternary ammonium head groups. The electrode was fabricated using a PtRu/C catalyst. A power density of 2200 mW cm^−2^ in an H_2_/O_2_ environment was obtained for a single unit cell. Ul Hassan et al. [61] synthesized poly(norbornene) tetrablock copolymer ionomers, the ionomers were incorporated into AEMFCs with symmetric and asymmetric electrode configurations. Using the said electrode, a long-term stability of 2000 h was achieved. Interestingly, the cell exhibited a peak power density of 3200 mW cm^−2^ under H_2_/O_2_ gas fueling.

The ionomer content has a prominent effect on controlling the water management in both the electrodes in AEMFCs [62]. For example, the higher ion exchange capacity (IEC) of the ionomers at the cathode resulted in a reduction in OH^−^ ionic resistance and thus improved cell performance, while the relatively lower IEC of ionomer at the anode avoids detrimental effects such as ionomer dissolution and anode flooding. Thus, it is imperative to design an ionomer with proper design consideration. Table 3 shows some commercial products currently used for electrode designs.

#### 2.4.2. Water Management and Transport Properties

Among the significant issues with AEMFCs, water management for the system and carbonation when exposed to CO_2_ are the factors that significantly affect the performance of AEMFCs. Water management in PEMFCs has been extensively explored, but the development of water management in AEMFCs were slow and these were extremely lacking in terms of their durability and ionic conductivity performance compared to PEM. However, as the advent of new processes and advances in research improve the durability of AEMs, the operation of water management becomes increasingly important. Similar to proton-conducting membranes, AEMs conduct OH^−^ ions in the presence of water [63]. As shown in Figure 4b AEMFC, the OH^−^ ions generated in the cathode travel through the AEM to the anode where they react with H_2_ fuel to produce H_2_O.

This reaction is accompanied by the release of electrons which pass through an external load to the cathode, generating electricity. The humidification or water content of the membranes dictates the mobility of the OH^−^ anion through the membranes. Consequently, it is pertinent to control the humidification level and water transport properties of the ion-conducting membranes. To maintain proper working conditions, liquid water transport should be effectively regulated; ineffective water management includes water flooding and dehydration. The former case leads to limited mass transport in the anode, whereas the latter case leads to the dehydration or even fracture of the membrane [65]. A dry membrane will result in poor OH^−^ conductivity, which will affect the reaction rate on the anode side. Moreover, the ohmic resistance of the cell will drastically increase. The membrane conductivity is approximately 300 times higher when fully hydrated, and thus maintaining a hydrated membrane ensures the sustained performance of the fuel cell. Pivovar et al. and Tschinder et al. [66,67] reported that the electro-osmotic drag coefficient (ξ), that is, the number of water molecules associated with each mobile ion, plays a significant role in the water management of AEMs. Normally, the movement of water inside the membrane occurs by diffusion owing to the difference in water activity on both sides of the membrane [68]. Continuous efforts have been devoted to understanding the movement of water in AEMs; however, owing to the poor stability of the membrane electrode assembly (MEA) combined with poor cell performance, research has been limited. Many researchers have postulated that the movement of water occurs from the anode side to the cathode side, which is the reverse of the process occurring in PEMFCs [69]. Recently, Zhang et al. [68] stated that the nature of the membrane also has a significant influence on water movement. They observed that for hydrophilic membranes, water movement occurs from the anode to cathode, whereas the opposite process occurs in hydrophobic membranes. Moreover, in AEMFCs, water plays different roles: at the cathode, water acts as a reactant, whereas it is a reaction product at the anode [70]. The dehydration and flooding of the cathode and anode, respectively, are more prominent in AEMFCs than PEMFCs, leading to poor cell performance. Slade and Varcoe [27] investigated this phenomenon. Varcoe [71] studied the influence of relative humidity on the water content and ionic conductivity of quaternized alkaline AEMs. In most instances, the behavior of water in ion-conducting membranes often influences the transport properties, especially in trade-off relationships between ion conductivity and molecular or ion permeability and permselectivity. In many cases, if higher conductivity is desired, more water must be absorbed into the membrane, which promotes larger molecular and ion diffusion coefficients [72]. The correlation between the water uptake, conductivity, and selectivity of these membranes is that the sizes of pores and channels increase as the water uptake increases, and as the size of the channels increases, the ion mobility and the conductivity of the membrane increases [73]. Decreased water absorption generally results in decreased transport properties, and ion permeability [74]. Huo et al. stated that the current density is the most influential factor for water distribution in the anode [65]. The higher current density results in a greater water production rate which might cause the blockage of the flow channels in the anode layer. Flooding can also occur under low-current-density conditions such as the high relative humidity of gas and low temperatures. In addition, excessive water absorption poses a potential risk (e.g., mechanical properties) of swelling of the AEM, reducing ion conduction and membrane stability. Therefore, it is important to maintain a balance between ionic conductivity and water absorption to prevent the flooding and drying of the membrane and electrode [73]. Analytical methods such as magnetic resonance imaging (MRI), neutron imaging, electron microscopy, and X-ray imaging are often employed in order to understand the state of water in fuel cells [75,76,77,78,79,80]. Various mathematical and numerical models were developed to study the water transport properties of AEMs in alkaline fuel cells [65,81].

#### 2.4.3. Contamination of AEM with CO_2_

One of the main reasons for the degradation of AEMFC performance is the difficulty of maintaining the ionic conductivity of the membrane due to the presence of CO_2_ contamination. When pure oxygen is fed to the cathode, OH^−^ is generated and acts as the charge carrier. However, when air is used instead of pure oxygen, a carbonation process occurs in which OH^−^ reacts with CO_2_ ((6,7) by Equation (5)): (5)$$O_{2}+2H_{2}O+4e^{-}\to 4{\mathrm{OH}}^{-}$$
(6)$${\mathrm{OH}}^{-}+{\mathrm{CO}}_{2}\;\to {\mathrm{HCO}}_{3}^{-}$$
(7)$${\mathrm{HCO}}_{3}^{-}+{\mathrm{OH}}^{-}\to {\mathrm{CO}}_{3}^{2}{}^{-}+H_{2}O$$

The main result of this carbonation process is a significant decrease in the anion conductivity (OH^−^) available for an electrode reaction in the AEM and consequently a reduction in the AEMFCs power density and durability. The carbonation process is reversible. Compared to the OH^−^ conductivity of AEM, the conductivity of CO_3_^2−^ and HCO_3_^−^ is low, and the transport mechanism is slow [82]. This is because of the larger size of the carbonate and bicarbonate ions than OH^−^. This can be mitigated through self-purging, etc., but may affect the chemical stability of the membrane and, in turn, the stability of the cell performance (as shown in Figure 5).

In [84], Yanagi et al. studied the concentration of anionic species after the exposure to ambient air was measured using A201 (a commercially available membrane composed of a hydrocarbon-based backbone and quaternary ammonium groups as an ion exchange site). The membrane has a fast CO_2_ gas absorption rate, and it was confirmed that the overall OH^−^ generated was converted to CO_3_^2−^ or HCO_3_^−^ within 30 min. It was suggested that the ionic conductivity due to the neutralization of CO_2_ dropped from 42 mS cm^−1^ to 10 mS cm^−1^ due to CO_2_ gas. Grew et al. [82] studied the correlation between ion conductivity and pH during the membrane exchange of OH^−^, CO_3_^−^, and HCO_3-_ according to temperature and CO_2_ content. A highly alkaline environment has been reported to decrease ionic conductivity due to nucleophilic attack and the Hoffman elimination mechanisms of AEM. The cause of the instability at high pH in AEMs is generally considered to be the substitution of cationic ammonium groups by hydroxide anions, which are good nucleophiles. The mechanism for degradation is presented as (1) attack via direct nucleophilic displacement and (2) anion exchange membrane attack via Hoffman removal via β-hydrogen sites. Studies have shown that durability can be increased, however, it is accompanied with a decrease in conductivity. However, Matsui et al. [85] injected CO_2_ into the anode, which resulted in a decrease in current density and ionic conductivity and an increase in pH due to the occurrence of overpotential in the anode. The limitation of power density due to the effect of CO_2_ has impeded the development of AEMFC to date. The power density of the AEMFC measured with a radiation-grafted AEM material was 823 mWcm^−2^ in a humidified H_2_/O_2_, 60 °C environment [86] Suzuki et al. [87] studied the effect of CO_2_ dissolution with AEM (A201, Tokuyama Co.) by controlling the concentration of carbonate ion species in the membrane and the CO_2_ concentration in the fueling gas. As the content of carbonate ion species decreased, the membrane conductivity increased. It was observed that the overpotential of the anode electrode and the cell performance decreased with the increase in the supply of CO_2_ to the cathode. This suggested that the formation of carbonate ion species near the three-phase boundary had a dominant effect on the overpotential of the electrode. Pandey et al. [88] compared the conductivity of Fumasep^®^ FAA membranes exposed to CO_2_ in a controlled environment. The ionic conductivity of the AEM decreased from 75 mS cm^−1^ to 17 mS cm^−1^ when exposed for 2 days at 95% RH and 80 °C; this decrease was due to the membrane clogging by HCO_3_^−^. Divekar et al. [89] reported a drop in OH^−^ concentration upon the air exposure of AEM and a loss of membrane water content following bicarbonate and carbonate conversion. Krewer et al. [90] presented a model for analyzing the effect of CO_2_ on AEM and AEMFC through the equilibrium results of the carbonation process. As a result of carbonation equilibrium, when OH^−^ generated by the electrochemical reaction is exposed to ambient air, the concentration of CO_3_^2−^ in the membrane increases. After a certain period, the concentration of CO_3_^2−^ decreases, forming HCO_3_^−^. It was suggested that CO_2_ from the ambient air enters the cathode during AEMFC operation, affects the anion composition inside the membrane and in the anode catalyst layer, and thus affects the fuel cell performance. The modeling carried out in this study confirmed that the carbonation process was not beneficial to the cell and that the current density and fueling flow rate at the cathode were the variables that affected the carbonate content of the membrane. They also reported that when operating the cell at a high current density, the effect of CO_2_ can be significantly reduced. Peng et al. [91] found that the mobility of anions was greatly affected by the conductivity of AEM and the interaction between functional groups in the polymer and water molecules. It has been reported that HCO_3_^−^ ion interferes with water transport by preventing water molecules from diffusing within the polymer. This was confirmed by observing the significant decrease in cell voltage and overpotential. Zheng et al. [83] reported that the voltage range was reduced by 0.2~0.4 V depending on fuel and CO_2_ during AEMFC operating conditions. The report suggested that the performance decrease was caused by the accumulation of carbonate ions in the cathode rather than the resistance caused by carbonation. In the operating AEMFC, the current density, operating temperature, and CO_2_ concentration control are cell operating factors for carbonation reduction. However, the problem is difficult to solve because carbonation is not an electrochemical reaction product. When operating the cell at a specific current density (>1 A cm^−2^), the suboptimal ‘self-purge’ of preventing voltage loss by CO_2_ is not a complete solution as it is time consuming. Zheng et al. [92]. reported that decreasing total the CO_2_ capacity (reducing cathode flow) and increasing hydration levels of in AEMFCs are valuable ways for lowering CO_2_-related voltages. As the anode fuel supply flow rate decreases, the anode CO_2_ concentration increases to a level greater than the cathode inlet concentration. In another study by Zheng et al. [93], they observed that in addition to affecting the conductivity of AEM, carbonation also has a detrimental effect on the polymer morphology which results in the decrease in the water content of the cell. For the membranes shown in Figure 6, the voltage loss trend by CO_2_ contamination is HDPE-BTMA > LDPE-BTMA > GT72-5 > PAP-TP-85 > GT64-15 > GT78-15. It was confirmed that carbonate transport and stabilization tended to be slower in GT78-15 and LDPE-BTMA AEM than in GT64-15 and HDPE-BTMA AEM. This suggested that GT-64-15 and HDPE-BTMA AEMs were CO_2_ tolerant. The carbonation of the membrane can be minimized using an AEM with high ionic conductivity and decarbonization during operation through a self-purging mechanism. Refueling through complete CO_2_ removal not only adds complexity to the system but also increases the cost of operation. Thus, ensuring a high conductivity of AEM, it can be presumed that the carbonation of the AEM due to CO_2_ can be effectively reduced.

### 2.5. Anion Exchange Membranes

#### 2.5.1. Requirements of the Membrane

As the membrane forms the core of the AEMFCs, it should conduct OH^−^ ions well, exhibit no electronic conduction and low fuel crossover, and retain its physico-mechanical properties in the hydrated state [94]. Failure to maintain stability will result in performance degradation. This is mainly due to the chemical degradation of ionomers and AEMs in the high-pH alkaline environment in the system. The attack of OH^−^ in an alkaline medium on the polymer used in the anion exchange membrane destroys the ionic properties of the polymer [95]. The high nucleophilicity and basicity of the OH^−^ ion can degrade the covalently bonded counteraction and the polymer backbone. Chemical degradation reduces the conductivity of the membrane and increases overall cell resistance, reducing durability.

The foremost task in designing a membrane is to consider its ionic conductivity (≥100 × 10^−3^ S cm^−1^) and its mechanical strength [96]. Enhancing the membrane conductivity by increasing the functional group density is often accompanied by a loss in mechanical strength. Additionally, thinner membranes result in zero gaps between the anode and the cathode, resulting in lower ohmic losses and improved power–density output. However, if it is made thin, the durability will be lowered. This is because the polymer structure is destroyed as the thin membrane absorbs moisture and dries, and problems such as fuel crossover occur. The polymer backbone has good mechanical strength, but the flexibility of AEM is not satisfactory. This results from them being brittle under conditions of low relative humidity, which causes problems during long-term fuel cell operation [97]. Consequently, the optimization of the membrane characteristics (e.g., crosslinking, functional group density, flexibility, etc.) is required to produce an efficient energy device. Table 4 shows AEM products prepared for commercialization.

(1)The hydrate ion conductivity of AEM, which can ensure high power density.(2)AEM with excellent chemical and mechanical stability in the AEMFC environment (temperature, humidity, alkali composition, etc.).(3)Easy to manufacture and inexpensive AEM.

#### 2.5.2. Membrane Design

Taking into account the relevant literature from the past decade, the synthesis of AEMs can be categorized into three major approaches:(1)Physical grafting: In this process, the prepared membrane is bombarded with gamma radiation to generate reactive radicals which are then allowed to react with a quaternary group containing reactive sites. This process allows the introduction of quaternary ammonium or phosphonium groups’ membrane structure. The physical grafting techniques have been utilized to develop AEM based on perfluorocarbon membranes, as demonstrated by Varcoe, Slade, and co-workers [27]. Although this approach is convenient and clean, it is unable to produce the ionomer solution. An ionomer is crucial in the fabrication of membrane electrode assembly (MEA) [98,99].(2)Chemical grafting: In this approach, quaternary ammonia functional groups are grafted onto the polymer chain through chemical reaction on pre-functionalized polymer with chloromethyl or bromomethyl groups. Since in this method the functionalized polymer is solubilized in certain organic solvents, ionomer solutions can be readily obtained. Moreover, this process allows obtaining a high degree of quaternary amine functionalized AEM allowing the development of high ion-conducting AEMs. This efficient, complete approach has been widely adopted by several research groups. Polysulfone (PS) and its analogs are the most widely used polymer backbones, mainly because PS is a commercially mature product with outstanding stability and is capable of forming a flexible thin-film with high mechanical strength [100,101,102].(3)Polymerization: In this process, AEM starts from quaternary-ammonia-containing monomers and is synthesized through polymerization reactions from monomers functionalized with quaternary amine groups. [103,104]. The type of AEM is considered to demonstrate tailorability and versatility in terms of the fabrication of the membrane as reported by Coates and co-workers [105,106]. This type of process of AEM fabrication can result in a membrane with very high ion-exchange capacity (IEC) and controlled molecular weight; however, the overall AEM performance—in particular, the mechanical strength and thermal stability—need further exploration.

### 2.6. Types of Anion Exchange Membrane

#### 2.6.1. Polyvinyl Alcohol (PVA)-Based Membranes

In this case, AEM results from the blending of an aqueous solution of PVA and a hydroxide salt. PVA is one of the most fascinating polymers and has been employed in numerous applications owing to its low cost, excellent mechanical properties, excellent solvent resistance, flexibility, and good adhesion to various surfaces [107]. PVA exhibits good film-forming capacity, good hydrophilic properties, and a high degree of reactive functional groups that are amenable to crosslinking using irradiation and thermochemical treatments. Its high dielectric constant and glass transition temperature are also deciding factors for fuel cell applications.

For PVA-based ion-exchange membranes, two important factors must be considered: the incorporation of ion-exchange components and the suppression of the PVA chain by swelling. The addition of acidic or basic components to PVA can address the former criterion [108,109], whereas crosslinking is generally carried out for decreasing the swelling characteristics [110,111].

Most of the early developments in PVA-based anion-conducting membranes utilized alkali metal hydroxides as the functional ionic source. Lewandowski et al. [112] developed PVA-based alkaline solid electrolytes by doping with KOH and water. They found that the ionic conductivity of the PVA–KOH–H_2_O polymer electrolyte (ca. 40 wt.% PVA, 25–30 wt.% KOH, and 30–35 wt.% H_2_O) was in the order of 10^−3^–10^−4^ S cm^−1^, and the activation energy (E_a_) was in the range of 22–28 kJ/mol. The ionic conductivity of the alkaline solid polymer electrolyte was dependent on the composition of KOH and H_2_O in the films. Yang and Lin [113] studied the ionic conductivity by increasing the KOH and H_2_O content. Although the membrane exhibited high ionic conductivity (47 × 10^−3^ S cm^−1^), they observed that the stability of the membrane deteriorated at higher temperatures.

Several approaches have been undertaken to tune the mechanical stability while simultaneously enhancing the ionic conductivity of PVA [114]. In an effort to increase the KOH retention capacity and the dimensional stability of PVA, Merle et al. [115] used poly(ethylene glycol) diglycidyl ether (PEGDGE) (Figure 7). The alkaline-fuel-cell test with this membrane demonstrated a power density of 72 mW cm^−2^ at 0.34 V and a high current density of 200 mA cm^−2^.

In another study, Wu et al. [116] synthesized a PVA/PAA (poly acrylic acid)/KOH solid polymer electrolyte system. The free radical polymerization of acrylic acid was carried out in the presence of PVA to form the alkaline polymer electrolyte. The membrane was then immersed in KOH solution, and ionic conductivity of 30 × 10^−2^ S cm^−1^ at room temperature (RT) was obtained for the PVA/PAA/KOH membrane [116]. Good mechanical properties and thermal stability were among this system’s attributes.

High alkaline stability in an aqueous medium was reported for interpenetrating poly(acrylic acid)-modified poly(vinyl alcohol) (PVAAA) membranes by Yang et al. [117]. The ionic conductivity was correlated with the PAA content: 60 wt.% PAA in PVA exhibited the highest ionic conductivity (0.312 S cm^−1^). For PVA, high ionic conductivity is achieved by striking a balance between the degree of crystallinity and water uptake. Such an interpenetrating structure decreases the degree of crystallinity, thereby facilitating the swelling of the membrane. The increase in the amorphousness promotes the faster transport of the ionic species in the polymer electrolyte [118].

Yang et al. [119] compared the ionic transport properties of PVA with those of a PVA/polyepichlorohydrin (PECH) blend and PVA/tetraethyl ammonium chloride (TEAC) system (Figure 8). The anionic transport number (*t*^−^) of the PVA/TEAC blend was found to be higher than that of the PVA/PECH blend; however, the latter exhibited more mechanical restraint. The higher *t*^−^ is explained by the fact that the OH^−^ ion and quaternary ammonium group interact extensively via coordination, enhancing the number of charge carriers.

By virtue of its pendant hydroxyl groups, PVA offers an opportunity to modify its structure. The modification of PVA by quaternized groups can significantly enhance the stability of these types of membranes. The fixing of quaternized groups onto the chains provides a unique opportunity to modulate the ionic conductivity as well as the mechanical durability of the PVA under humidified conditions. Quaternized ammonium groups were directly introduced into the main chain of the PVA backbone using a chemical grafting technique followed by crosslinking with glutaraldehyde (Figure 9) [120].

Qiao et al. developed a PVA-based membrane by incorporating poly(acrylamide-co-diallyldimethylammonium chloride) (PAADDA) as an anion charge carrier [121]. The OH^−^ conductivity of the membrane increased with an increase in the PAADDA content in the PVA matrix and the temperature, reaching 0.74–12 mS cm^−1^ with direct crosslinking and 0.66–7.1 mS cm^−1^ with indirect crosslinking in the temperature range of 30–90 °C. Qiao et al. fabricated a PVA membrane by blending PVA and PVP, followed by chemical crosslinking and KOH doping. The membrane displayed high stability and was found to be stable up to 120 °C in aqueous 10 M KOH [122]. Ye et al. [123] attempted to incorporate fixed charges on the PVA backbone by modifying PVA with 4-methyl-4-glycidylmorpholin-4-ium chloride (MGMC) and crosslinking with the diglycidyl ether of bisphenol A (DGEBA). The ionic conductivity of this type of membrane was found to be 5.21 × 10^−2^ S cm^−1^ at 60 °C. Hari Gopi et al. [124] functionalized PVA with quaternary ammonium groups using hexadecyltrimethylammonium bromide (HDT). The distribution of QA groups in the polymer increased with the content of HDT, a conductivity of 5.7 mS cm^−1^ at 40 °C was obtained for PVA-HDT with 15 wt.% HDT. A conductivity loss of approximately 10% and 21% after the immersion of the AEM in 4 M KOH after 150 h and 300 h, respectively, was reported, indicating sufficient stability. However, these stability results for PVA-based AEM were not significantly encouraging; thus anion-conducting interpenetrating polymer networks (IPNs) were evaluated as AEM to compensate for the problems encountered with PVA. Through an IPN structure, the synergistic effect of the two constituent polymers can be exploited and balanced properties in areas such as ionic conductivity, mechanical properties, and stability can be obtained [125]. Zuo et al. [126] developed IPNs based on PVA and polyethyleneimine (PEI). The hydroxide ion conductivity of the 1:1 prepared membrane in the study was 4.87 mS cm^−1^ at 80 °C and 5.83 mS cm^−1^ at 90 °C. Table 5 lists the ionic conductivity of each PVA-based membrane.

#### 2.6.2. Polysulfones (PS)

The introduction of an aromatic moiety into the polymer backbone is thought to be one of the best ways of developing high-performance AEMs. In particular, AEMs are more susceptible to degradation or loss of mechanical performance under hydrated conditions than PEMs. Thus, the mechanical properties of the membrane play a pivotal role during operation, as its integrity will influence the lifetime of the fuel cell. Many high-performance polymers have been synthesized and used in fuel cell operations, namely poly(arylene ether)s such as poly(arylene ether sulfone) and poly(arylene ether ether ketone) (PEEK). The origin of PS dates back to as early as 1965 [127,128] when Union Carbide first introduced it to the market. However, these polymers were limited to use in specific applications. Compared to other polymers, PS has the highest service temperature performance. Furthermore, its high chemical inertness makes it suitable for application in fuel cell membranes [129]. One of the biggest advantages of PS is the ease with which it can be fabricated into a membrane with reproducible properties.

Owing to their high chemical stability, PS membranes (Figure 10) have attracted widespread applications in domains such as electrodialysis, nanofiltration, separation technique, and AEM in fuel cells [130,131,132,133,134,135]. Zschocke and Quellmalz [136] demonstrated that a PS membrane exhibits excellent chemical properties when soaked in 40 wt.% aqueous solutions of NaOH at 70–80 °C for over 300 h. Owing to its high stability in an alkaline medium, AEMs based on PS were developed. The stability of the cationic head group is important in this regard; under the operating conditions (i.e., high pH and high temperature), the degradation of the head group often occurs [137,138,139].

Owing to the high chemical and physical stability exhibited by the PSU, anion exchange groups were introduced into its backbone following chloromethylation and subsequent quaternization. The PS membrane exhibited an IEC of 0.5 and 1.8 × 10^−3^ eq g^−1^ and electrical resistances between 0.5 and 5 Ω cm^−2^ [136]. As AEMs have low ionic mobilities compared with those of PEMs, membranes with reduced thickness are a reasonable solution. Wang et al. recently synthesized a composite membrane based on polytetrafluoroethylene and 1,4-diazabicyclo-[2.2.2]-octane (DABCO)-functionalized PS (QDPSU). The membrane was 30 µm thick and demonstrated a mechanical strength of 32 MPa. Moreover, an ionic conductivity of 0.05 S cm^−1^ at 50 °C was achieved [140]. Such thin membranes with good mechanical attributes can significantly enhance the performance of the fuel cell. Rao et al. [141] fabricated imidazolium-functionalized block copolymers of PS with well-separated hydrophilic and hydrophobic phases at the molecular level which was attributed to good hydrolytic stability under swelled conditions. AEMs of this type were prepared by introducing quaternary ammonium groups via chloromethylation followed by quaternization. Such membranes possess a high ionic conductivity but simultaneously demonstrate higher swelling characteristics which lower the mechanical stability. Over the years, quaternized ammonium PSs have been a suitable choice for fabricating AEMs for alkaline-fuel-cell applications. High ionic conductivity can be obtained by manipulating the nanophases of the polymer segments [142]. Nafion, a well-known PEM, demonstrates nanophase separation resulting in the generation of nano-ionic channels through which water molecules can move in coordination [143]. Nafion is composed of a polytetrafluoroethylene (PTFE) backbone and pendant short perfluoroether branches terminated with sulfonic acid groups. Hydration after water adsorption leads to the nanophase phase separation of the hydrophobic matrix, hydrophilic clusters, and sulfonated domain aggregates to form ionic clusters. Then, the formed clusters are interconnected to create nano-ion channels through which ions can pass [144,145]. Li et al. carried out an amination of PS via lithiation [146], achieving a high degree of substitution without significantly affecting the polymer structure. Gong et al. [144] quaternized polysulfone (QPSF) and poly(vinyl alcohol) and then form an IPN structure. The ionic conductivity of QPSF alone was 25.2 mS cm^−1^, which was lower than that of the IPN-based AEM of QPSF/PVA of 7:3 (*w/w*) (23.4 mS cm^−1^) at 60 °C. Furthermore, the ionic conductivity increased to 18.2 mS cm^−1^ with a higher PVA content (6:4, *w/w*). Moreover, the author stated that the higher the PVA content, the higher the mechanical and alkaline stability. Teresa Pérez-Prior et al. [145] reported a quaternized PSU with 1,4-diazabicyclo [2.2.2]octane(DABCO) chloromethylated polysulfone (CMPSU). The authors reported that the DABCO group confers stability to the polymer mainly because it is impossible to achieve the semi-peripheral confirmation required to have Hofmann removal. Thus, two nitrogens in DABCO structure can prevent the decomposition of the polymer by the stabilization of the positive charges. Ureña et al. [147] functionalized PSU with three different ionic groups (tetramethylammonium, 1-methylimidazolium, and 1,2-dimethylimidazolium) and crosslinked with N,N,N′,N′-tetramethylethylenediamine (TMEDA). They have shown that the hydrophobic segments in the network improved the thermal, durability, and dimensional stability without reducing ionic conductivity. In addition, the membrane functionalized with 1-metyylimidazolium had the highest ionic conductivity. Table 6 provides the ionic conductivities of the PS-based membranes discussed.

#### 2.6.3. Polyphenylene-Based Membranes

Poly(2,6-dimethyl-1,4-phenylene oxide) (PPO) possesses excellent chemical, thermal, and mechanical properties. Lin et al. synthesized a series of uncrosslinked and crosslinked poly(phenylene oxide) (PPO)-based AEMs containing bis-imidazolium cations [148]. The authors observed that uncrosslinked AEMs had a higher swelling and water uptake than the crosslinked AEMs. The imidazolium-PPO AEM exhibited a conductivity of 23.00 mS cm^−1^ at 20 °C, which rose to 54.04 mS cm^−1^ at 90 °C. The H_2_/O_2_ single-cell test with a crosslinked membrane (Cross-Im-PPO-75) produced a power density of 212.45 mW cm^−2^. Pan et al. [149] prepared a series of AEMs containing two side-chain alkyltrimethylammonium groups with n-propyl (C3) and n-pentyl (C5) alkyl chains attached onto the PPO main chain (i.e., NC_3_Q-PPO, NC_5_Q-PPO, and NC_5_ Q-PPO). It was observed that after the AEMs were immersed in 1 mol L^−1^ NaOH solution at 80 °C for 30 days, the NC3Q-PPO, NC_5_Q-PPO-40, and NC_5_Q-PPO-60 membranes showed conductivity retentions of 73.1%, 89.9%, and 81.2%, respectively, compared with 39.8%, 41.2%, and 56.5% for the QPPO-40, QC_3_-PPO-40, and QC_6_-PPO-40 samples, respectively. The AEMs NC_5_Q-PPO-40 and NC_5_Q-PPO-60, having longer spacers between the aromatic polymer chains, not only displayed good stability but also very high ionic conductivities of 73.9 and 96.1 mS cm^−1^ at 80 °C, respectively. In addition, the swelling ratios of the AEMs were restricted to 15% and 28%. Xue et al. [150] investigated a single- and dual-cation AEM based on PPO incorporating spirocyclic 3,6-diazaspiro [5.5] undecane (DSU) cation/cation strings. The PPO-based single- and double-cation AEMs (PPO-SDSU and PPO-DDSU, respectively) possessed well-connected hydrophilic domains that resulted in an increase in the ionic conductivity of the AEMs. Thus, the PPO-SDSU membrane with an IEC of 1.91 meq g^−1^ exhibited a hydroxide conductivity of 31.9 mS cm^−1^ and a linear dimensional swelling of 9.6% at 20 °C. Carlson et al. [151] synthesized the AEMs of poly (phenylene oxide)-membranes with a five-carbon alkyl spacer between the backbone and a trimethylalkylammonium (TMA) or piperidinium (Pip) cationic group. The AEMs demonstrated IECs between 1.5 and 1.9 mequiv g^−1^ for TMA and Pip, respectively. The authors reported that the introduction of a five-carbon alkyl spacer did not enhance the performance; furthermore, exchanging the TMA with a Pip cationic group resulted in poorer fuel cell performance but a higher IEC. Lee et al. [152] prepared different PPO-based AEMs with quaternary ammonium groups of varying side-chain lengths. They observed that when the length is similar, the crosslinked membranes exhibited higher alkaline stability compared to their uncrosslinked counterparts. In this study, the crosslinked cQPH with a hexyl acyl chain showed an anion conductivity of 105 mS cm^−1^ at 80 °C owing to a good phase separation. Li et al. [153] explored the effects of a side-chain architecture on the membrane performance of dual-side-chain-grafted poly(phenylene oxide) AEMs and found that the AEM possessed a high ionic conductivity of 21.3 mS cm^−1^ at 30 °C when the cation’s central atom was attached to a long hydrophobic extender and an additional hydrophilic side chain (next to the cation). In addition, such AEMs were observed to exhibit good alkaline and dimensional stability and a high retention of conductivity (90.5%) after 1 M NaOH treatment at 60 °C for 528 h. Additionally, the incorporation of a tricationic and hydrophobic side chain enhanced the ionic conductivity of the resulting AEM to 50.0 mS cm^−1^ at 30 °C; and the ionic conductivity retention after 1 M NaOH treatment at 60 °C for 528 h was as high as 90.6%. Table 7 summarizes the ionic conductivity of the polyphenylene-based membrane.

#### 2.6.4. Ionic Liquid-Based Membrane

Ionic liquids (ILs) are a novel class of organic electrolytes that are generally molten at ambient temperature. ILs are environmentally benign and possess outstanding physiochemical properties and excellent thermal and mechanical properties. Furthermore, they demonstrate good conducting properties and possess wide electrochemical windows [103,104,154]. To obtain novel AEMs with unique properties such as high conductivity and stability at elevated temperatures with good chemical and mechanical attributes, ILs have been used as AEMs for fuel cell applications. The first studies on IL-based AEMs were reported by Guo et al. [104] and Lin et al. [105]. In their respective studies, AEMs based on imidazolium cationic groups were synthesized that exhibited excellent alkaline stability under high-pH conditions at 60 °C. Ye and Elabd [155] showed that the homopolymer of an imidazolium-based IL (PIL) remained chemically stable over a broad pH range in different alkaline solutions; they attributed this to the delocalization of the imidazolium cationic charge through conjugation. On the contrary, the PIL-based homopolymer was not promising as an AEM owing to its high water solubility. This issue was overcome by using PIL-based block copolymers, wherein a hydrophobic moiety acting as a mechanically reinforcing block was introduced into the polymer backbone, resulting in a robust water-insoluble membrane [155,156,157]. Qiu et al. [156] synthesized a novel AEM by the in situ crosslinking of a bis-imidazolium-based IL monomer, 1-allyl-3-(6-(1-butyl-2-methylimidazol-3-ium-3-yl)hexyl)-2-methylimidazol-3-ium bromide ([ABMHM][Br]_2_), with styrene and acrylonitrile using UV irradiation (Figure 11). The hydroxide conductivity reached up to 2.5 × 10^−2^ S cm^−1^ for the AEM having a 40% mass fraction of [ABMHM][OH]_2_. In addition, the AEM displayed excellent long-term stability in an alkaline solution at 60 °C. The author credited the high alkaline stability of [PABMHM]_40_[OH]_2_ to the resonance effect of the conjugated imidazole rings.

Fang et al. [157] copolymerized two types of ILs containing an imidazolium moiety (i.e., 1-vinyl-3-methylimidazolium iodide [VMI]I and 1-vinyl-3-butylimidazolium bromide [VBI]Br). The ionic conductivity was reported as approximately 22.6 mS cm^−1^ at 30 °C. Interestingly, the membranes were stable in 10 M NaOH at 60 °C for 120 h without experiencing a significant deterioration of the ionic conductivity. Additionally, the fuel cell performance evaluated using this membrane in a H_2_/O_2_ single unit showed an open circuit voltage (OCV) of 1.07 V, whereas the peak power density of 116 mW cm^−2^ was achieved at a current density of 230 mA cm^−2^ at 60 °C. Ouadah et al. prepared IL-functionalized copolymers via the radical polymerization of poly(butylvinylimidazolium) (b-VIB) with para-methylstyrene (p-MS), forming b-VIB/p-MS block copolymers [158]. They subsequently crosslinked the block copolymers with poly(diphenylether bibenzimidazole) (DPEBI) to yield a PIL-based AEM. The poly(butylvinylimidazolium) functioned as the conducting block, whereas the DPEBI moieties resulted in high membrane stability. The crosslinked b-VIB/p-MS displayed a hydroxide conductivity of 35.7 mS cm^−1^ at 25 °C, whereas the hydroxide conductivity reached 73.1 mS cm^−1^ at 100 °C. Table 8 provides the ionic conductivity for each ILs-based membrane.

#### 2.6.5. Polyolefin-Based Membranes

##### Radiation Grafting

It is broadly accepted that AEMs such as polyolefins and polyphenylenes exhibit higher alkaline stability owing to their heteroatom-free all-carbon backbones [159]. Additionally, properties such as hydrophilicity, microphase separation, flexibility, and mechanical strength can be regulated by adjusting AEM backbones [160,161]. Recently, the development of AEMs has made a significant leap. This includes polyolefin-based AEMs in general, and specifically, radiation-grafted AEMs (RG-AEMs), incorporating low-density polyethylene (LDPE) films produced using a high-dose electron-beam, which have dramatically advanced fuel cells. In the latter case, high-energy irradiation (from a 60 Co γ-ray or electron beam) of the polymer generated active sites such as free radicals or peroxides on the surface of polyolefin materials; these active sites were then grafted with p-chloromethyl styrene and polymerized to quaternize the polyolefins. These types of AEMs in AEMFCs demonstrated high H_2_ fuel-cell performances (>1.0 W cm^−2^ at >60 °C with non-Pt cathodes) [162,163,164]. This exceedingly good performance was attributed to LDPE RG-AEMs with high ionic conductivities, stability, and fast water-transport characteristics.

A radiation-grafting modification has been used as an important tool for simply and efficiently functionalizing polymers post-polymerization. In the fabrication of AEMs, polyolefins are often subjected to this procedure [165]. Danks et al. [166] prepared AEMs using γ-ray-irradiated polyvinylidene fluoride (PVDF) and tetrafluoroethylene–hexafluoropropylene copolymer (FEP). The irradiated polymer membranes were immersed in a grafting solution of p-chloromethyl styrene, and grafting rates of 54% and 27% were obtained for the AEMs based on PVDF and FEP, respectively. The grafted polymers were then quaternized with trimethylamine solution to obtain the AEM. Mamlouk et al. [86] reduced the relationship between the radiation-grafting conditions of the three polyolefins (ETFE, LDPE, and high-density polyethylene) and compared the performances of the respective AEMs. A high grafting rate equivalent to that of hydrocarbon polymers is conducive to improving the conductivity of the membrane. Wang et al. [167] further optimized the conditions of radiation pretreatment and graft polymerization by using water as a dilution solvent instead of isopropanol, which was previously used. This method not only reduced the radiation dose of the electron beam (from 70 to 30–40 kGy) and the concentration of p-chloromethyl styrene (from 20% to 5% *v*/*v*) but also produced an AEM with a high IEC (2.10 mmol g^−1^).

From the aforementioned preparation methods, it can be seen that the functional monomer copolymerization and polymer grafting methods to prepare polyolefin AEMs have their own advantages and disadvantages. The cost of functional monomers is relatively high, and complex separation and purification techniques are required to increase the membrane efficiency.

##### Direct Polymerization

Generally, ring-opening metathesis polymerization (ROMP) and Ziegler–Natta-catalyst-mediated polymerization techniques have been successfully employed to synthesize polyolefin AEMs by using functional monomers such as α-olefin, norbornene, and cyclooctene [168,169,170,171,172,173]. Therefore, the chemical structures and compositions can be readily tuned to achieve the optimized properties of the resulting AEMs. In recent years, norbornenes and cyclooctenes have been among the most studied functional monomers (Figure 12) [174,175,176].

Through ROMP, an AEM material with a polyolefin main-chain was obtained by the Coates research group [175,176]. Using a DA reaction, they designed and synthesized a quaternary-ammonium-salt-containing norbornene functional monomer (1) without β-H1 using a Grubbs second-generation catalyst. A subsequent ROMP with dicyclopentadiene resulted in a crosslinked AEM. The conductivity test results of the membrane showed that when the IEC was 1.0 mmol g^−1^ at 20 °C, the hydroxide-ion conductivity was 14 mS cm^−1^, the swelling rate in methanol was 1.7%, and the tensile strength of the film reached 16 MPa.

Coates et al. [168] then used quaternary-ammonium-salt-based cyclooctene functional monomer 3 to prepare the respective crosslinked polyolefin AEM membrane with high conductivity. Zha et al. [177] designed and synthesized divalent norbornene-based ruthenium-ion functional monomer 4, and through ROMP with dicyclopentadiene, obtained a crosslinked metal-ion polyolefin AEM. However, the polyolefin obtained using the ROMP method contained a large number of double bonds in the main chain, resulting in a less stable main-chain structure under alkaline conditions. Therefore, in order to prepare alkaline-stable AEM materials, in a subsequent study, the Coates research group [174,176] used a metal-catalyzed hydrogenation method to obtain organic-solvent-soluble polyolefin AEM materials, such as that depicted in Figure 13. Using the same method, Zhu et al. [178] prepared a soluble polyolefin AEM material containing a cobalt cation. The ionic conductivity of the described polyolefin-based membrane is shown in Table 9.

#### 2.6.6. Organic–Inorganic-Based Membranes for AEMFC

In order to develop more efficient AEMs, significant research effort has been devoted to exploring novel membrane materials. In this regard, organic–inorganic polymer nanocomposites have been extensively studied for their potential application as membranes in fuel cells [179,180,181]. However, most of these studies focused on PEMFCs [182]. The combination of nanoscale materials with an organic polymeric matrix endows PEMs with the advantages of rigidity, high thermal stability, and good mechanical attributes. The high surface area of the nanoscale material leads to good interfacial interaction with the polymer matrix (Figure 14) [181,183,184]. Das et al. [185] studied hybrid organic–inorganic anion-conducting membranes prepared via 1,4-diglycidyl butane ether (DGBE)-aided chemical grafting of silica (SiO_2_) nanoparticles onto PVA. The ionic conductivities of these composite membranes were determined to be in the range of 10^−4^–10^−3^ S cm^−1^ under 100% relative humidity [186]. In another work, Das et al. developed an anion-conducting composite membrane by crosslinking hydroxide-conducting 1,4-diazabicyclo-[2.2.2]-octane (DABCO)–cellulose nanofibers with DABCO–PS using 1,4-dibromo butane. Interestingly, the composite membranes exhibited very high ionic conductivities in the range of ca. 39–74 mS cm^−1^ at 25 °C and reached 128 mS cm^−1^ at 80 °C. The high surface area of the cellulose nanofibers combined with the tailored quaternization of cellulose significantly enhanced the ion-exchange capacity; furthermore, a well-defined microphase separation and interconnected ionic channels were attributed to the increased ionic conductivity [187]. Das et al. also performed the chemoselective quaternization of chitosan using DABCO as the bi-cationic amine, in which the quaternized chitosan was amalgamated with 1,4-diazoniabicycle-[2.2.2]-octane-functionalized PS to obtain crosslinked AEMs. Ionic conductivities of 54 and 94 mS cm^−1^ were obtained at 25 and 70 °C, respectively, for a membrane containing 2 wt.% quaternized chitosan (DMC). Moreover, when the 2 wt.% DMC composite membrane was applied to a urea/O_2_ fuel cell, a peak power density of 4.4 mW cm^−2^ and current density of 16.22 mA cm^−2^ at 70 °C were obtained [188]. This result indicated that composite membranes could provide a low operation cost and high performance, making them promising for commercialization. However, the long-term stability of the fuel cell in a real-world scenario was not evaluated in the study, and thus, further research is still needed [189]. In order to enhance the ionic conductivity as well as the chemical and mechanical stability, Das et al. fabricated a hybrid composite membrane incorporating quaternized chitosan and graphene oxide with DABCO as the filler in a PPO matrix. The AEM with a quaternized GO/quaternized cellulose/quaternized PPO weight ratio (*w/w*) of 1/1/100 displayed a hydroxyl conductivity of ≈114 mS cm^−1^ at 25  °C and ≈215 mS cm^−1^ at 80  °C. Jiang et al. [190] developed a novel quaternized PVA/chitosan/molybdenum disulfide (QPVA/CS/MoS_2_)-based AEM, and Miao et al. [191] synthesized quaternized polyhedral oligomeric silsesquioxanes (QPOSS)/quaternized PS (QPSU)-based AEM. In this study, the QPOSS was synthesized by reacting octaammonium POSS (OAPOSS) with glycidyltrimethylammonium chloride (GDTMAC) via an epoxide ring-opening reaction. The QPSU-3%-QPOSS demonstrated an ionic conductivity of 53.6 mS cm^−1^ at 80 °C. Quaternized poly(arylene ether sulfone) (QPAES)/nano-ZrO_2_ composite membranes were developed by Li et al. [192] The QPAES/nano-ZrO_2_ membranes containing over 7.5 wt.% nano-ZrO_2_ possessed ionic conductivities exceeding 41.4 mS cm^−1^ at 80 °C. Table 10 shows the ionic conductivity of the organic-inorganic-based membranes described above.

## 3. Conventional Ion-Conducting Membranes: Pros and Cons

In AEMs, the mobile charges are hydroxide ions rather than protons [194]. In recent years, considerable efforts have been devoted to the development of AEMs with enhanced properties [195]. In order to be useful in alkaline fuel cell applications, the requirements for AEMs must be stringent; they should possess high ionic conductivity, an efficient barrier to the fuel gases, high stability of the cationic head group, and durability under hydrated conditions. The electrokinetic advantage of alkaline fuel cells incorporating AEMs were discussed in several publications [196,197]. The main shortcoming of AEMs is the degradability of the quaternized head group in alkaline environments.

## 4. Conclusions

This review comprehensively summarized the different types of AEMs derived from different backbone materials and functional groups. As mentioned, by using AEMs in fuel cells, some common issues such as fuel crossover could be eliminated. Furthermore, AEMs are also advantageous because of their ability to operate in alkaline environments, meaning less expensive catalysts can be used. Moreover, AEMs can contribute to the diversification of liquid fuels for AEMFCs by allowing the use of small organic molecules due to the low crossover. Moreover, AEMFC offers a clean and efficient conversion device among others. In the last decade, many researchers have reported power densities comparable to PEMFCs. Table 11 shows the commercialization requirements that must meet the protocols proposed by the Department of Energy for commercial standardization. Although it has an advantage from the catalytic point of view, it is necessary that the fuel cell achieves higher durability and output through systematic analysis and tread-off rather than focusing on the content corresponding to one aspect as a system. Based on these set targets, the following suggestions for future AEMFC development are given below.

The development of an anion exchange membrane material that can ensure chemical stability and long durability under high pH conditions must (2) focus on cost reduction and affordability that can boost AEMFC’s development. (3) An anion exchange membrane must be developed that can lower the loss of membrane conductivity due to CO2 poisoning. (4) Flooding and drying out must be prevented through the development of efficient water management strategies for the hydration of anion exchange membranes. (5) Systemic deterioration due to carbonate accumulation in the cathode must be alleviated. (6) Efficient catalyst that can secure economic feasibility must be developed (development of a non-platinum catalyst that can have the advantage of catalysis in an alkaline environment).

AEMFC research and its commercialization must necessarily be realized through extensive collaboration among researchers around the world with expertise in different aspects of fuel cell development. Research in the future should focus more on AEM properties under real AEMFC working conditions and on improving AEMFC preparation. The lifespan of a polymer-based fuel cell is greatly affected by AEMs. The focus should be on solving the major problems of stability at high pH in terms of anion exchange membranes, the hydration of intact membranes from little water, and carbonates in the carbonation process. As a result, it is possible to overcome the chronic limitation of the high cost of PEMFC and achieve the zero-emission goal of the international community in the near future. A cleaner, green Earth that secures energy sources through sustainable development and creates a beautiful society for future generations is within reach.

## Figures and Tables

**Figure 1 polymers-14-01197-f001:**
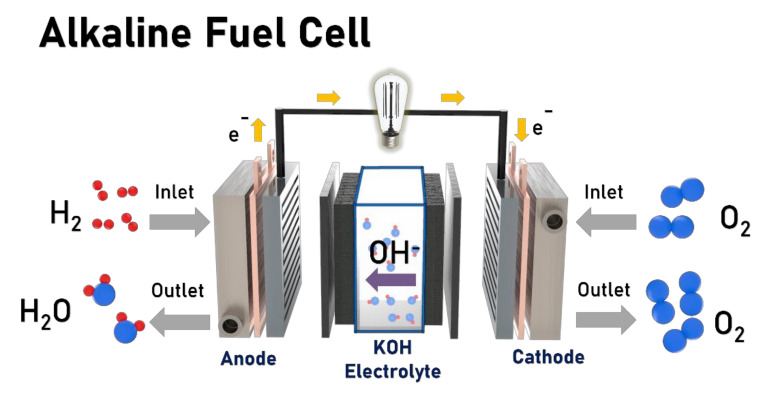
Schematic diagram of AFC.

**Figure 2 polymers-14-01197-f002:**
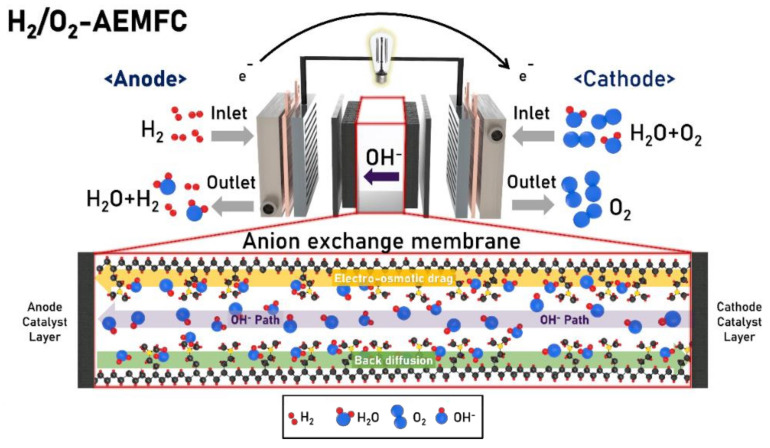
Schematic diagram of the H_2_/O_2_ AEMFC.

**Figure 3 polymers-14-01197-f003:**
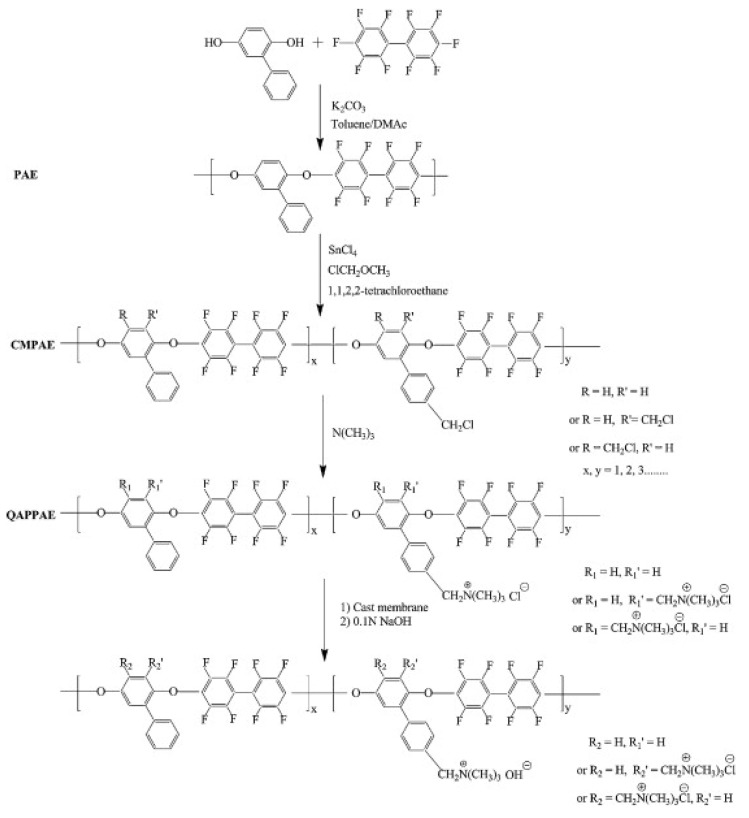
Synthetic route for partially fluorinated copoly(arylene ether) ionomers with pendant quaternary ammonium groups (Reprinted with permission from ref. [54] Copyright 2011 Elsevier.).

**Figure 4 polymers-14-01197-f004:**
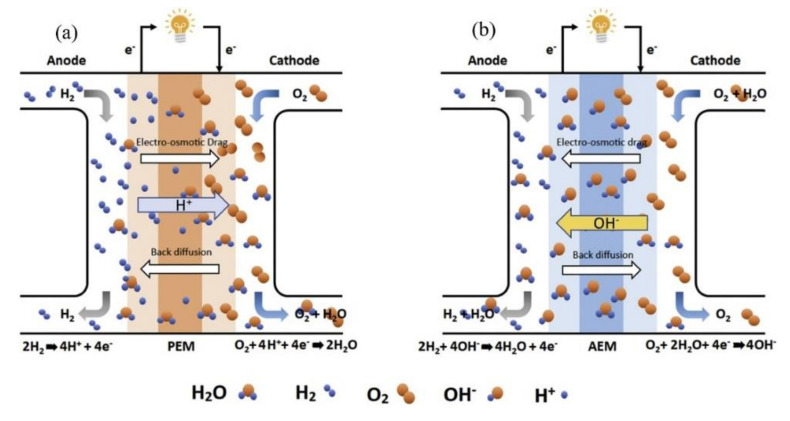
Schematic representation of (**a**) PEMFC and (**b**) AEMFC (Reprinted with permission from ref. [64] Copyright 2020 Elsevier).

**Figure 5 polymers-14-01197-f005:**
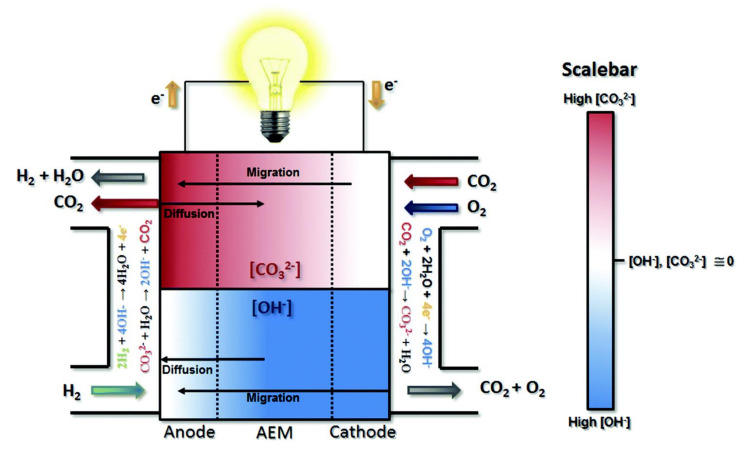
Schematic diagram of carbonate and hydroxide transport and distribution in an AEMFC operating with CO_2_ present in the cathode reactant gas. The upper section of the diagram isolates the CO_3_^2^ behavior of the operating cell with a color gradient representing the concentration gradient. The upper section of the diagram shows the OH^−^ concentration gradient and hydroxide migration and diffusion direction (Reprinted with permission from ref. [83] Copyright 2019 Royal Society of Chemistry).

**Figure 6 polymers-14-01197-f006:**
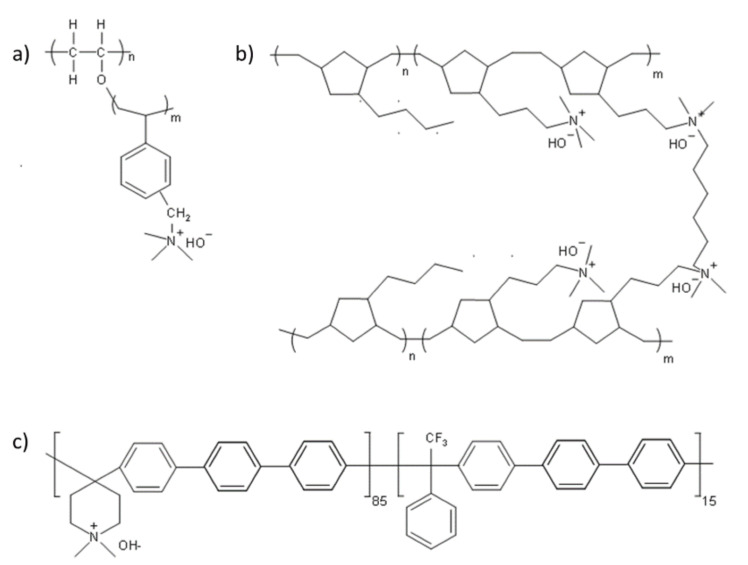
The structure of anion exchange membranes (AEMs) used in this study (**a**) high-density polyethylene polymer with a benzyltrimethylammonium cation (HDPE-BTMA) and low-density polyethylene polymer with a benzyltrimethylammonium cation (LDPE-BTMA), (**b**) poly (norbornene) copolymer of GT64-15, GT72-5 and GT78-15 and (**c**) poly(aryl piperidinium)-based polymer with a terphenyl chain (PAP-TP-85). (Reprinted with permission from ref. [93] Copyright 2021 MDPI.).

**Figure 7 polymers-14-01197-f007:**
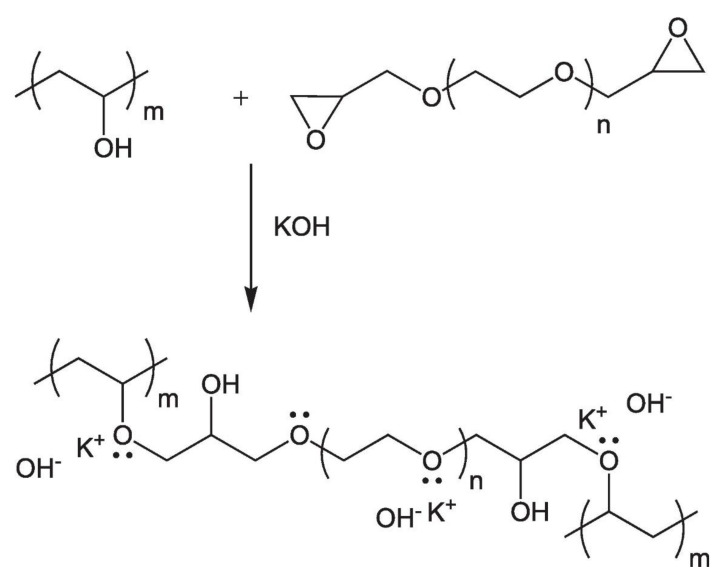
Schematic representation of the synthesis of the AEMs based on PVA, KOH, and PEGDGE (Reprinted with permission from ref. [115] Copyright 2012 Elsevier).

**Figure 8 polymers-14-01197-f008:**
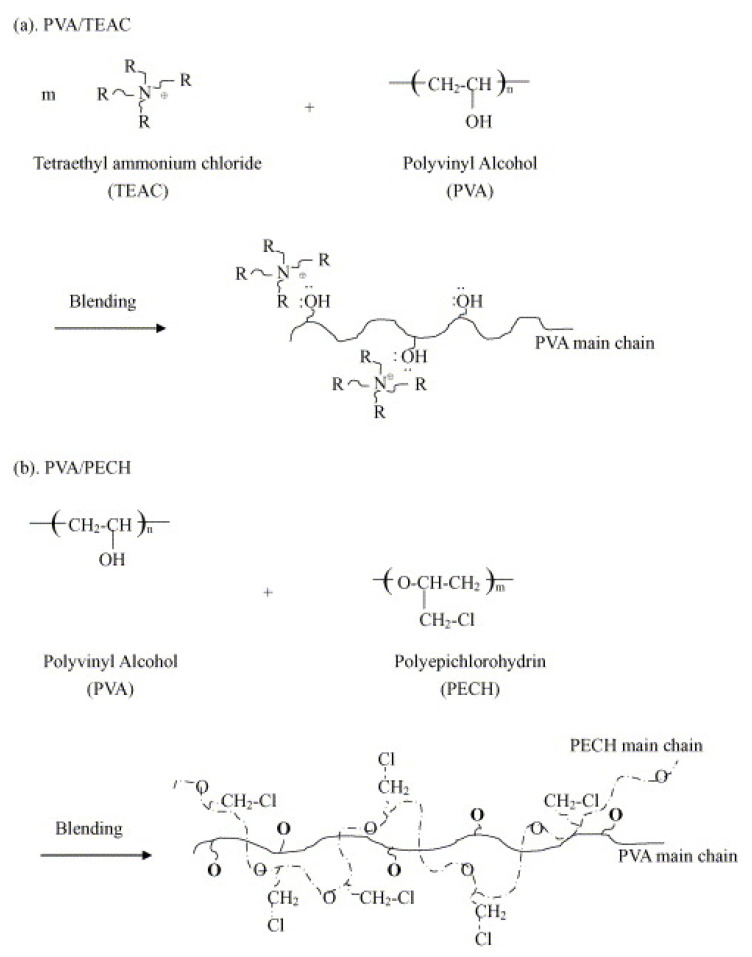
Preparation of the PVA polymer blended with (**a**) TEAC and (**b**) PECH (Reprinted with permission from ref. [119] Copyright 2005 Elsevier).

**Figure 9 polymers-14-01197-f009:**
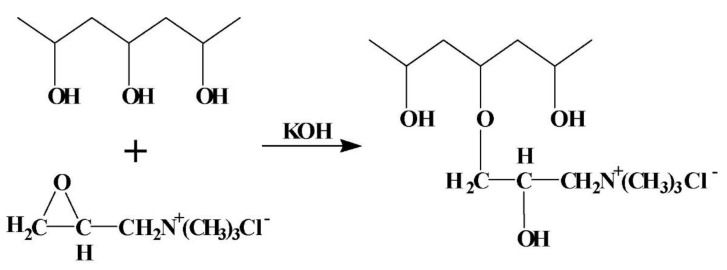
Chemical grafting of a quaternized group onto the PVA main chain (Reprinted with permission from ref. [120] Copyright 2008 Elsevier).

**Figure 10 polymers-14-01197-f010:**
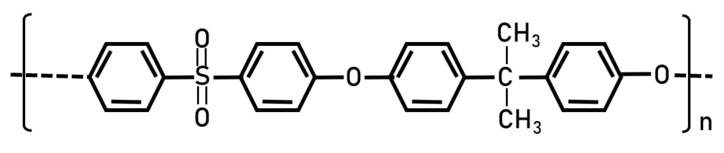
Structure of PS.

**Figure 11 polymers-14-01197-f011:**
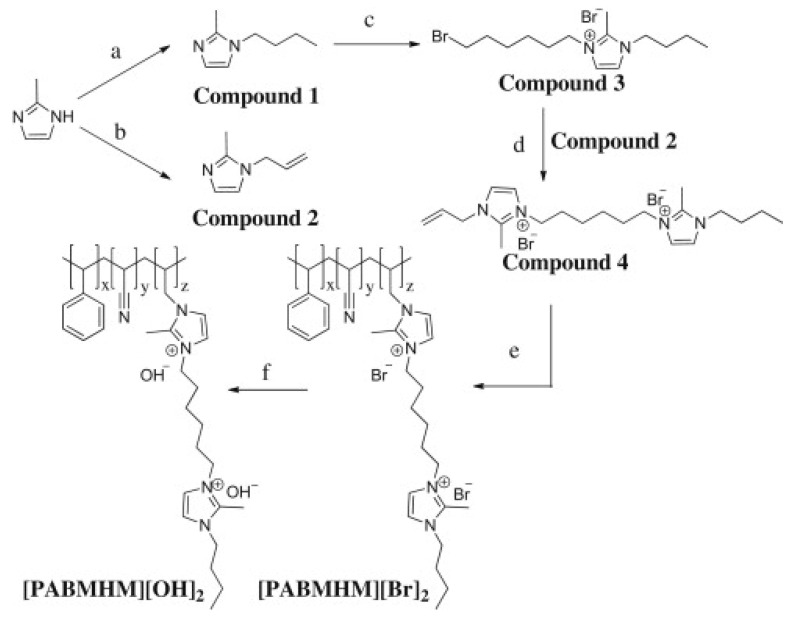
Synthetic procedure for AEMs: (**a**) 1-bromobutane, NaOH, RT, 4 h; (**b**) allyl bromide, NaOH, RT, 4 h; (**c**) 1, 6-dibromohexane, 50 °C, 2 d; (**d**) hydroquinone, 50 °C, 2 d; (**e**) styrene, acrylonitrile, DVB, photo-crosslinking, 0.5 h; and (**f**) 1 M KOH, 60 °C, 24 h (Reprinted with permission from ref. [156] Copyright 2012 Elsevier.).

**Figure 12 polymers-14-01197-f012:**
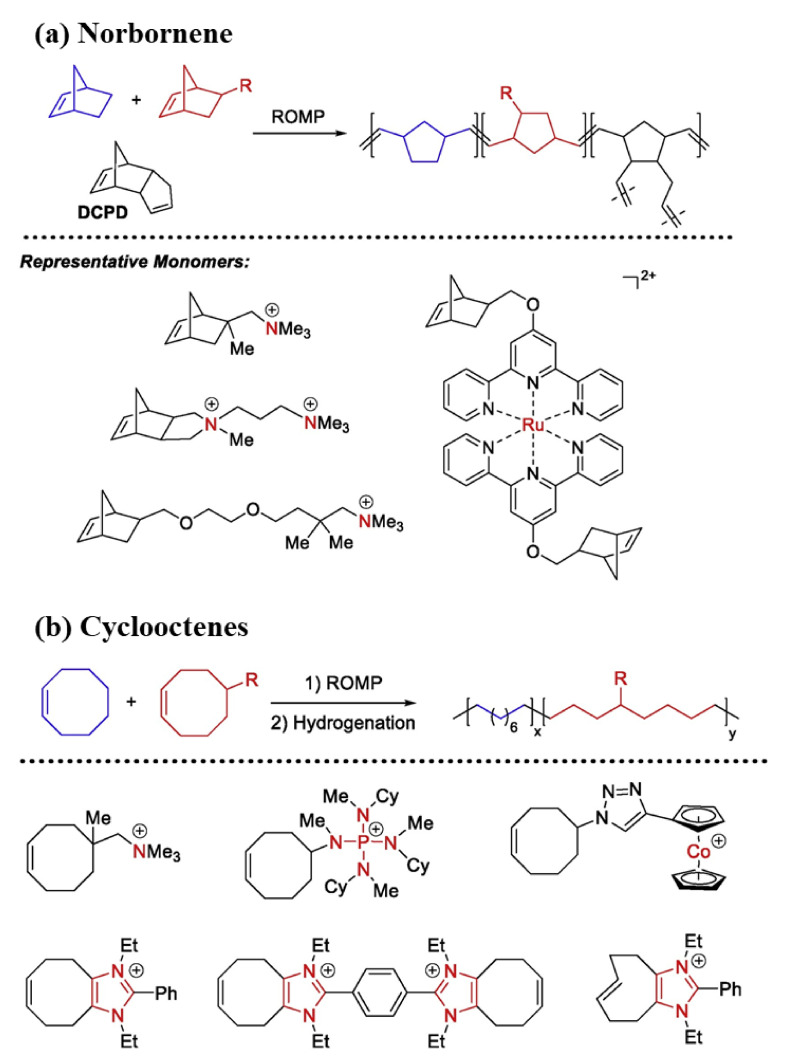
Direct polymerization via the ROMP of (**a**) norbornene; (**b**) cyclooctenes derivatives (Reprinted with permission from ref. [176], Copyright 2018 American Chemical Society).

**Figure 13 polymers-14-01197-f013:**
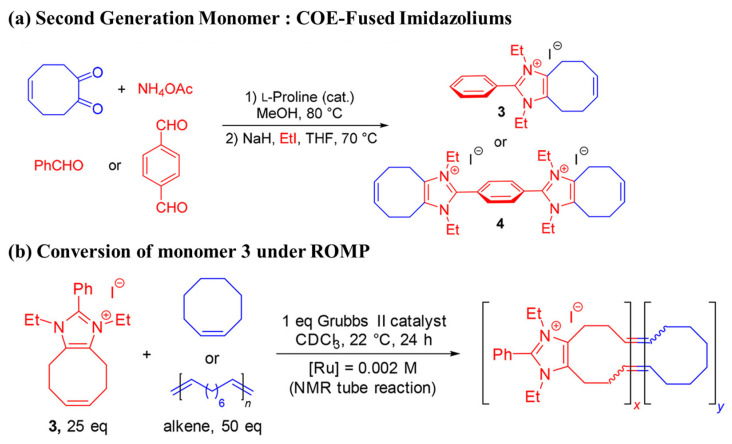
Synthesis of (**a**) imidazolium monomer 3; (**b**) polyolefin AEM materials (Reprinted with permission from ref. [174] Copyright 2018 American Chemical Society).

**Figure 14 polymers-14-01197-f014:**
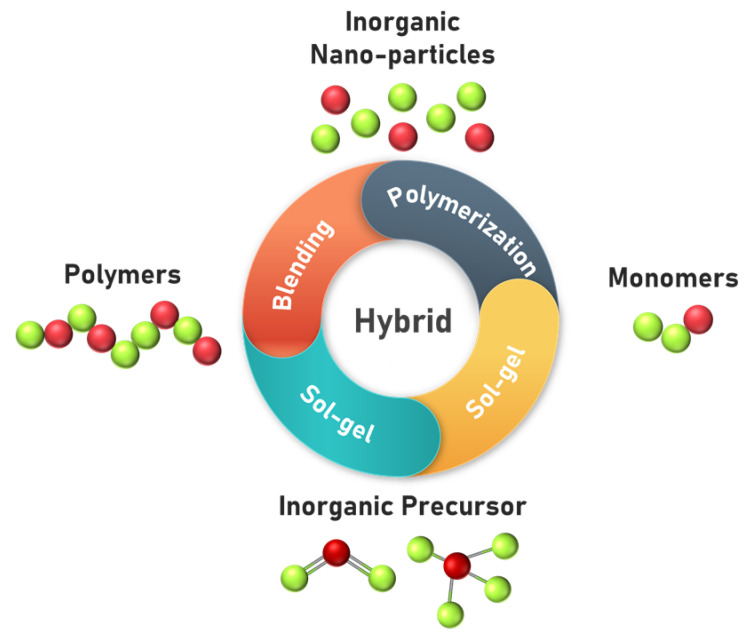
Schematic representation of the different preparation routes for organic–inorganic hybrid materials and AEMs.

**Table 1 polymers-14-01197-t001:** Table of the properties of lithium-ion (LIB), lead-acid and vanadium redox flow batteries (VRFB) [9].

Type	Energy Density (Wh/L)	Power Density (W/L)	Nominal Voltage (V)	Life Cycle	Depth of Discharge (%)	Round Trip Efficiency	Estimated Cost (USD/kWh)
Li-ion	200–400	1500–10,000	4.3	10,000	95	96	200–1260
Lead-acid	50–80	10–400	2.0	1500	50	82	15–475
VRFB	25–33	1–2	1.4	13,000	100	70	315–1050

**Table 2 polymers-14-01197-t002:** Ion conductivity property of membrane by the manufacturer.

Type	Membrane Brand Name	Manufacturer, Country	Thickness (µm)	Conductivity (mS/cm)	Ref.
Cation	Nafion 117	Dupont, USA	183	76.5 ± 1.7	[45]
	Nafion 212	Dupont, USA	50.8	157.0 ± 2.1	
	Gore-select^®^	Gore, USA	18	100	[46]
	Gore-select^®^	Gore, USA	35	96	
Anion	Fumasep^®^ FAA-3	Fumatech, Germany	45–50	40–45	[47]
	Tokuyama A201	Tokuyama, Japan	28	42	
	Sustainion^®^ 37–50	Dioxide Materials, USA	50	70	

**Table 3 polymers-14-01197-t003:** Commercially available ionomers for electrode design.

Manufacturer	Brand/Commercial Name
Surrey	SION1
Tokayama, Japan	AS-4 anion exchange ionomer
Fuma, Germany	Fumion/FAA-3-SOLUT-10
Xergy, USA	Pention™/Pention-D72
Dioxide Materials, USA	Sustainion^®^/XB-7

**Table 4 polymers-14-01197-t004:** Commercial anion exchange membranes and their suppliers.

Membrane/Manufacturer	Structure	IEC (meq g^−1^)	Thickness (mm)	Resistance (cm^2^)
Tokuyama Co., Ltd., Tokyo, Japan	PS/DVB	1.4–1.7	0.12–0.18	2.0–3.5
RAI Research Corp., Hauppauge, NY, USA	LDPF	0.9	0.24	4.0–7.0
CSMCRI, Bhavnagar, India	LDPE/HDPE (IPN)	0.8–0.9	0.16–0.18	2.0–4.0
Solvay S.A., Bruxelles, Belgium	Morgane ADP	1.3–1.7	0.13–0.17	1.8–2.9
PCA Polymerchemie Altmeier GmbH, Heusweiler, Germany	PC 100 D	1.2 quat.	0.08–0.1	5

**Table 5 polymers-14-01197-t005:** Ionic conductivity of PVA-based membranes.

Membrane	Thickness (µm)	Conductivity (mS/cm)	Condition	Ref.
PVA (43.3 wt.%)–KOH (35.7 wt.%)–H_2_O (21 wt.%)	800	0.13	25 °C	[112]
PVA (30 wt.%)–KOH (40 wt%)–H_2_O (30 wt.%)	480	47.1	30 °C	[113]
PVA (60 wt.%)–polydiallyldimethylammonium chloride (PDDA)	110	22	25 °C	[114]
PVA (5 wt.%)–KOH (10 wt.%)–PEGDGE (5 wt.%)	250 ± 50	220 ± 3	25 °C	[115]
PVA (10 wt.%)–PAA (7.5 wt.%)–KOH (32 wt.%)	450	301	25 °C	[116]
PVAAA (PVA60 wt.%, AA40 wt.%)–KOH (40 wt.%)	-	312	25 °C	[117]
PVA(50 wt.%)-PVC(50 wt.%)	150–200	540.1	25 °C	[118]
PVA–KOH	480	47.2	25 °C	[119]
PVA/PECH (1:1) blend	530	1	25 °C	
PVA/TEAC (1:1) blend	460	23.1	25 °C	
Crosslinked QAPVA	-	7.34	30, in DI water	[120]
PVA/PAADDA	90–110 (by composition mass)	0.74–12	30–90 °C, in DI water	[121]
PVA/PVP/KOH-d (PVA/PVP, 1:0.5 by mass)	60–80	530	25 °C, in 8M KOH solution.	[122]
Crosslinked and quaternized poly(vinyl alcohol) (CLQPVA)	-	52.1	°C, in DI water (RH 100%)	[123]
QAPVA-hexadecyl trimethylammonium bromide (HDT) (15%)	-	4.84	30 °C, in DI water	[124]
PVA/PEI	-	4.87	80 °C	[126]

**Table 6 polymers-14-01197-t006:** Ionic conductivity of PS-based membranes.

Membrane	Thickness (µm)	Conductivity (mS/cm)	Condition	Ref.
PTFE-quaternary 1,4-diazabicyclo-[2.2.2]-octane (DABCO) polysulfone (PTFE-QDPSU) composite membrane	30	51	55 °C, in DI water, RH 100%	[140]
Imidazolium-functionalized PES membrane (EI-PES)	25–30	100	80 °C, in DI water	[141]
Ammonium-functionalized membrane (QA-PES)		73		
QPSF-PVA (7:3 wt.%)	-	25.2	60 °C,	[144]
QPSF-PVA (6:4 wt.%)		18.2		
DABCO-functionalized polysulfones membrane (PSU-DABCO–OH 58%)	115	0.157	25 °C, in 0.1 M KOH solution	[145]
Crosslinked DABCO-functionalized polysulfones membrane (C–PSU–DABCO–OH 133%)		0.167		
Phenyltrimethylammonium functionalized polysulfone membrane (PSf-PTMA)	-	58	80 °C	[146]
functionalized PSU (degree of crosslinking 20%, crosslinked polymer/PSU: 6/4)	-	0.01	25 °C, in 0.1 M KOH solution	[147]
sIPN-MIm–OH functionalized PSU (degree of crosslinking 15%, crosslinked polymer/PSU: 9/1)	-	0.09		
sIPN-DMIm–OH functionalized PSU (degree of crosslinking 5%, crosslinked polymer/PSU: 6/4)	-	0.04		

**Table 7 polymers-14-01197-t007:** Ionic conductivity of polyphenylene-based membranes.

Membrane	Thickness (µm)	Conductivity (mS/cm)	Condition	Ref.
Bis-imidazolium based poly(phenylene oxide) membrane	-	54.04	90 °C, in DI water	[148]
Crosslinked bis-imidazolium based poly(phenylene oxide) membrane	-	Approximately 32		
NC_4_Q-PPO-40	50 ± 3	73.9	80 °C	[149]
NC_5_Q-PPO-60		96.1		
PPO-SDSU-36	50 ± 5	51.6	60 °C, in DI water	[150]
PPO-DDSU-27		47.2		
PPO5-TMA-1.9	30 ± 5	110	60 °C in DI water (RH 90%)	[151]
PPO5-Pip-1.8	45 ± 5	93		
Hexyl acyl chain and a crosslinked cQPH	-	105	80 °C, RH 100%	[152]
Dual-grafted mono-quaternized PPO 16C25-3O25	90	21.3	30 °C	[153]
Dual-grafted tri-quaternized PPO 3QA16C16-3O16		50		

**Table 8 polymers-14-01197-t008:** Ionic conductivity of ILs-based membranes.

Membrane	Thickness (µm)	Conductivity (mS/cm)	Condition	Ref.
AmimCl:MMA = 6:1 (mole ratio)	33.1	33.3	30 °C, in DI water	[104]
[PVMIm][OH] _40_ -DVB _2_	50	55.8 ± 5.5	60 °C, in DI water, RH 100%	[105]
Poly(MEBIm-OH)	80–200	9.6	30 °C, RH 90%	[155]
[PABMHM]_40_[OH]_2_	45	25	90 °C	[156]
[VBI]Br:Styrene = 1:1.5 Feed ratio of (IILs/Styrene, mole)	45	22.6	30 °C, in 0.1M/L NaOH solution	[157]
Crosslinked b-VIB/p-MS	-	35.7	25 °C	[158]

**Table 9 polymers-14-01197-t009:** Ionic conductivity of polyolefin-based membranes.

Type	Membrane	Thickness (µm)	Conductivity (mS/cm)	Condition	Ref.
Radiation grafting	Polyethylene-based membrane (PE-g–PVBC–TOH)	85–95	47.5	90 °C, in DI water	[165]
	F1NOH (FEP-g-PVBTMAOH membranes, g = 25.6%)	90–100	10–20	25 °C	[166]
	F2NOH (FEP-g-PVBTMAOH membranes, g = 22.7%)	75–82			
	Aminated poly(LDPE-g-VBC, g = 68%)	40	180–320	20–80 °C, RH 100%	[86]
	Aminated poly(HDPE-g-VBC, g = 47–56%)	40	140–280		
	Aminated poly(ETFE-g-VBC, g = 28%)	25	5–20		
	ETFE-based RG-AEM(30 kGy)	47 ± 2	68 ± 3 (Cl^−^)	80 °C, RH 100%	[167]
Direct polymerization	Imidazolium-fused cyclooctene monomer was prepared and subjected to ROMP applied to AAEM-1	–	29 ± 2	22 °C, in DI water	[174]
	Imidazolium-fused cyclooctene monomer was prepared and subjected to ROMP applied to AAEM-2	–	37 ± 2		
	Random copolymer from a trans-cyclooctene–fused imidazolium monomer (HC–[1]_498_[2]_200_)	–	134	80 °C	[175]
	ROMP of a bis(terpyridine)ruthenium(II) complex-functionalized norbornene 4 based membrane: DCPD (1:5 ratio)	103 ± 5	28.6	30 °C	[177]

**Table 10 polymers-14-01197-t010:** Ionic conductivity of organic–inorganic-based membranes.

Membrane	Thickness (µm)	Conductivity (mS/cm)	Condition	Ref.
PVA/DGBE-15/SiO_2_-5	140	7.14	25 °C, in DI water	[185]
QPSfQC15	-	128	80 °C, in DI water	[186]
Dimethyl chitosan crosslinked polysulfone QPSfDMC2	-	54.15 ± 2.10	25 °C, in DI water	[187]
Quaternized chitosan was amalgamated with 1,4-diazoniabicycle-[2.2.2]-octane-functionalized PS to obtain crosslinked membrane	-	151	80 °C, in DI water	[188]
Quaternized chitosan and graphene oxide with DABCO as the filler in a PPO matrix GO/cellulose/PPO (1/1/100 wt.%)	-	215	80 °C, in DI water	[189]
Quaternized poly(vinyl alcohol)/chitosan/MoS_2_ composite—QPVA/CS/MoS_2_-0.2	150	31.53	25 °C	[190]
Quaternized polyhedral oligomeric silsesquioxanes (QPOSS) into the quaternized polysulfone (QPSU) membrane QPSU-3%-QPOSS	30–45	53.6	80 °C	[191]
Quaternized poly(arylene ether sulfone)/nano-ZrO_2_ composite (ZrO_2_ content more than 7.5%)	-	>41.4	80 °C	[192]
Loading of the quaternized cellulose in the quaternized PPO (qPPO) matrix: qPPO/DG-Cel7 (7 wt.% of cellulose functionalized with DG)	-	164	80 °C, in DI water	[193]

**Table 11 polymers-14-01197-t011:** Organized by DoE, the target strategy for each year is given for the standardization of AEMFC [198].

Type	Proposed Milestones
2022	AEM fuel cell initial performance 0.65V at 1000 mA cm^−2^ on H_2_/O_2_ (maximum pressure of 1.5 atm) in MEA with total < 0.2 mg_PGM_ cm^−2^ and < 10% voltage degradation over 1000 h, T > 80 °C
2023	CO_2_ tolerance: < 65 mV loss for steady-state operation at 1.5 A cm^−2^ in H_2_/air scrubbed to 2 ppm CO_2_
2024	Catalyst durability: H_2_/CO_2_-scrubbed air after accelerated stress test < 40% loss after 10,000 cycles from 0.6 V to 0.95 V Membrane durability: 1000 h open circuit voltage hold at 70% RH and ≥80 °C
2025	1 W cm^−2^ at 0.65 V; H2/CO2-free air with total PGM loading < 0.125 mg cm^−2^. T > 80 °C, P ≤ 250 kPa
2030	AEM fuel cell peak power performance > 600 mW cm^−2^ under H_2_/air (maximum pressure of 1.5 atm) in PGM-free MEA
Ultimate	1 W cm^−2^ at rated power (~0.65 V at 95 °C), PGM-free MEA, T ≥ 80 °C, P ≤ 250 kPa

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
