# Peer review of "Anion Exchange Membranes for Fuel Cell Application: A Review"

_polymers, 2022, doi:10.3390/polym14061197_

Round 1

Reviewer 1 Report

In this manuscript, the development of AEMs for the AEMFC applications is comprehensively summarized since AMEFC can employ low platinum or non-platinum catalysts and have a low ORR activation energy. However, AEM is typically less durable and conductive than CEM. AEMFC is a promising technology that has reached technological saturation and vanquished technical limitations. This review focuses on various aspects of AEMs.

I consider the content of this manuscript will definitely meet the reading interests of the readers of the Polymers journal. Therefore, I suggest giving a minor revision and the authors need to clarify some issues or supply some more data to enrich the content.

  1. Abstract and Introduction
  • For the Keywords, ‘energy storage’, ‘non-platinum catalyst’, ‘power density’ and ‘low cost’ should also be added to attract a broader readership and highlight the superiority of AEMs.
  • Please pay attention to grammar problems, especially the missing or redundant definite articles. I suggest double-checking. I will point out several examples, but unfortunately, I cannot point out all of them. For example, in the Abstract, ‘The core of ensuring the power density of AEMFCs is the anion exchange membrane (AEM)’; Page 2, ‘Studies on AEMFCs are increasing to an alternative to these problems, continuously’; Page 7, ‘This process allows for the introduction of quaternary ammonium or phosphonium groups membrane structure’; Page 17, ‘The electrokinetic advantage of alkaline fuel cells incorporating AEMs has been discussed in several publications [158,159]’, and so on.
  • Page 2, ‘Among the various potential alternative energy options, fuel cells offer the most promising solution [1].

So how about renewable energies, such as wind and solar energy? For fuel cells, where does the hydrogen come from? I consider the delivery of hydrogen is also troublesome, and one possibility is to combine fuel cells with renewable energy generators, since ‘most of the renewable energy sources are intermittent, opening spatial and temporal gaps between the availability of the energy and its consumption by the end-users. Hence, it is necessary to develop suitable energy storage systems to be integrated with renewable energy generators to tackle the intermittent issue’ [Electrochimica Acta 309 (2019): 311-325]. So the surplus renewable energy can be used to produce hydrogen, which will be powered by fuel cells when the renewable energy production is interrupted.

  • Page 2, ‘Fuel cell technologies offer the advantage of direct conversion of chemical energy into electrical energy with the generation of heat and water [3]. Fuel cell exhibits an efficiency as high as 60% and, with >90% reduction in harmful pollutants [4]’.

   The above-described advantages indeed can be realized by many secondary batteries, and fuel cells are not the only choice. The efficiency of 60% is also not very high. In the introduction part, I suggest adding some comparisons about device properties between PEMFC&AEMFC with the most well-known secondary batteries, such as lithium-ion batteries (potential flammable and explosive risks), lead-acid batteries (large self-discharge, short service life), and flow batteries (low energy density and floor area is large), including efficiency, cost, lifetime, and energy density in a table briefly. [Advanced Materials 30.33 (2018): 1800561; Journal of energy storage 15 (2018): 145-157; Journal of Power Sources 493 (2021): 229445]

  • Page 2, ‘...maintaining the sustainability of fuel cell system components based on the use of platinum group metals (PGM)-based catalysts, application of high corrosion-resistant metal bipolar plates, and perfluorosulfonic acid (PFSA) such as Nafion membranes [6],[8],[9].

   For catalysts, the issue is easy to understand. But why can the application of high corrosion-resistant metal bipolar plate not be maintained sustainability? Is the bipolar plate very expensive or due to other reasons? This should be further clarified. And for membrane, it is not the use of ‘PFSA’ (as an acid), but the use of PFSA resins/polymers/membranes. In addition, the authors need to uniform AEMFC’ or ‘AMFC’ in the whole manuscript to be identical everywhere.

  1. Overview of Fuel Cells
  • Page 4, ‘However, the AEM employed in AEMFC ionic conductivity is lower than the commercial-grade proton exchange membrane (PEM).It is suggested to give an approximate conductivity range here to provide readers with a more intuitive comparison. In addition, ‘However, the biggest advantage of the alkaline fuel cell is that there is minimal fuel crossover’. But why does PEM have higher fuel crossover while AEM does not? The fuel normally is not an ionic compound. The mechanism differences of the fuel crossover through AEM and PEM should be clarified as well.
  • In the text, there are no sentences referring to Figure 1, nor is there any discussion and explanation of Figure 1. The appearance of Figure 1 is very abrupt. It is suggested to supplement the description for Figure 1. The same applies to Figure 11. And why in the caption of Figure 11, ‘Schematic representation of different preparation routes for organic-inorganic hybrid materials and PEMsappears PEMs?
  • Page 5, ‘The electrodes were prepared using 5 wt% (DMF) of poly(arylene ether sulfone) functionalized with quaternary ammonium groups.This description is incredible. Electrodes are generally carbon-based electrodes, not polymers. This should describe how the membrane is prepared by ionomer. Or the polymer is used as a binder like the function of Nafion. But in this case, the descriptions should also be modified.
  • Page 6, ‘In most instances, the behaviour of water in ion-conducting membranes often undermines the transport properties, especially in tradeoff relationships between ion conductivity and molecular or ion permeability and permselectivity.I consider here should not use undermines, but influences (not sure the effect is always negative). And the detailed mechanism of water to influence other properties should be explained better. If water uptake is reduced, generally the transport properties are reduced, especially the decline of ion permeability; while if the water uptake is higher, it may facilitate the transportations of all species [Electrochimica Acta 378 (2021): 138133].
  • Page 7, ‘Also, thinner membranes result in zero gaps between the anode and cathode, resulting in lower ohmic losses and improved power-density output.Do thinner membranes also bring about the drawbacks of higher crossover of fuels or active species? These disadvantages should also be mentioned.
  • Page 11, ‘Nafion, a well-known PEM, demonstrates nanophase separation resulting in the generation of nano-ionic channels, through which water molecules can move coordinately [104].How is the nano-phase separation formed? The difference of water absorption effect for hydrophilic and hydrophobic domains/clusters should be mentioned [Electrochimica Acta 378 (2021): 138133].

Author Response

원고를 검토하고 조언을 업데이트할 수 있게 해 주셔서 감사합니다.

첨부파일을 참조하시기 바랍니다.

Reviewer 2 Report

Current review manuscript entitled “Development of anion exchange membranes: A review” by Das et al focused on Anion exchange membrane fuel cells and Anion exchange membrane fuel cells. Manuscript written well and can be accepted after addressing the following comments

  1. In the conclusions section, Provide the challenges facing with the Anion exchange membrane fuel cells and Anion exchange membrane fuel cells.
  2. Comment on the usage of PVDF-HFP in the Anion exchange membrane fuel cells and Anion exchange membrane fuel cells
  3. Provide the future perspectives of the work.
  4. Improve the image quality of figure 2.
  5. Provide the table for the section 2.6. so that it will be easy for the readers to understand.

Author Response

Thank you for reviewing the manuscript and allowing us to update your advice.

Reviewer 3 Report

The review article titled "Development of anion exchange membranes: A Review" describes the development of anion exchange membranes in the last two decades. As a reviewer, I feel scared to state that the study seems to fail to connect the knowledge gap between the past and future of the development of the AEMFCs. However, in the last three years, significant improvements have been made in AEMs and AEMFCs. Why the authors have not included those reports in this study is not clear to me. This is a review that makes this report faulty or less effective for a broad spectrum of readers. So I would like to suggest the authors include the most recent reports with a more insightful evaluation.

  1. In the introduction and the AEMFCs overview, and through out the manuscript, the authors mix up AFC and AEMFC, which is confusing to the reader. The storyline should connect to the timeline with a certain difference between these two technologies. There are several general statements that leave a negative impression on the reviewer, and it appears that the authors are unfamiliar with these AFCs and AEMFCs. The authors should rectify the raised issue.
  2. In the types of anion exchange membrane, the authors have explained some old-fashioned polymers (heteroatom-containing polymers) and reported which ones should be avoided. In recent years, AEMFC enthusiasts have focused on polymers with non-heteroatom backbones such as SEBS, Polyphenylene, Polyolefine, and others. So, the authors should keep track of the current trend with the remaining challenges and prospects.
  3. While using specific cell diagrams schematically, the caption should also be specific instead of general. For example, PEM is regarded as a polymer electrolyte membrane; PEMFC is regarded as a proton exchange membrane fuel cell; AEMFC is regarded as an anion exchange membrane fuel cell. In figure 1, the figure caption should change the terms "PEM" and "PEMFC" to "AEM" and "AEMFC."
  4. In the challenges of AEMFC operation, the authors have stated "PEMFCs yields a power density of >500 mW/cm2 at 0.7 V under H2/air conditions, whereas analogous AMFCs provide a power density of <200 mW/cm2 under similar conditions." what might not be correct in the current context of the newly developed AEMFCs in the past few years. Authors should review some recent AEMFC-based articles which show comparable cell performance to the PEMFCs. For example, see recent publications: doi.org/10.1002/anie.202013395; doi.org/10.1149/2.1301910jes; doi.org/10.1021/acsapm.0c01405; doi.org/10.1002/aenm.202001986.
  5. The presented Figure 3 and the statement on Page 5 below Table 1 conflict with each other. The author should rectify the issue before resubmitting.
  6. In the challenges of AEMFC operation, it was expected to provide more insightful suggestions regarding the remaining challenges, which are missing. The authors should fill the knowledge gap for a broad spectrum of readers.
  7. The authors have used several lump citations, which should be avoided.
  8. The reference should be updated with the most recent reports.

Author Response

(The authors gave the same response as above.)

Round 2

Reviewer 3 Report

Since the authors have resolved most of the raised questions, I would like to request that the subject editor consider publishing the manuscript. Furthermore, I would like to remind the authors of my previous comments that in the last three years, significant improvements have been made in AEMs and AEMFCs. Why the authors did not include most of those reports in this study is not clear to me. This is a review that makes this report faulty or less effective for a broad spectrum of readers.